# Deep Learning Algorithms for Mean Field Optimal Stopping in Finite Space and Discrete Time

## Abstract

Optimal stopping of stochastic processes is a fundamental problem in optimization that has found applications in risk management, finance, economics, and recently in the fields of computer science. We extend the standard framework to a multi-agent setting, named multi-agent optimal stopping (MAOS), where a group of agents cooperatively solves finite-space, discrete-time optimal stopping problems. Solving the finite-agent case is computationally prohibitive when the number of agents is very large, so this work studies the mean field optimal stopping (MFOS) problem, obtained as the number of agents approaches infinity. We prove that MFOS provides a good approximate solution to MAOS. We also prove a dynamic programming principle (DPP), based on the theory of mean field control. We then propose two deep learning methods: one simulates full trajectories to learn optimal decisions, whereas the other leverages DPP to compute the value function and to learn the optimal stopping rule with backward induction; both methods train neural networks for the optimal stopping decisions. We demonstrate the effectiveness of these approaches through numerical experiments on 6 different problems in spatial dimension up to 300. To the best of our knowledge, this is the first work to study MFOS in finite space and discrete time, and to propose efficient and scalable computational methods for this type of problems.

## 1 Introduction

Optimal stopping (OS) has emerged as a powerful approach to tackle real-world problems characterized by uncertainty and sequential decision-making, in which the goal is to find the best time to stop a stochastic process (see Shiryaev (2007), Ekren et al. (2014)). Famous real-world OS problems are the job search (also called house selling or secretary problem) (Lippman and McCall, 1976) and in machine learning, the question of when to stop training a neural network can also be viewed as an instance of OS problems (Wang et al., 1993; Dai et al., 2019).

The OS framework has been extended to cover multi-agent scenarios, in which the aim is to stop several (possibly interacting) dynamical systems at different times in order to minimize a common cost function. We will refer to this setting at multi-agent optimal stopping (MAOS). MAOS has gained significant importance in a variety of fields. In robotics, for example, this theory has found applications in mission monitoring task, where multiple robots must observe the progress of other robots performing a specific task (Best et al., 2018). In finance the problem of pricing options with multiple stopping times (see Kobylanski et al. (2011)) can be viewed as an MAOS problem and was a motivation for Talbi et al. (2023; 2024).

However, the problem's complexity increases drastically with the number of agents. To tackle this issue, mean-field approximations can be used. In the case of multi-agent control, this leads to the theory of Mean Field Control (MFC), which aims to approximate very large systems of interacting agents who are cooperatively minimizing a social cost by choosing optimal controls; see (Bensoussan et al., 2013; Carmona and Delarue, 2018). Applications include crowd motion (Achdou and Lasry, 2019), flocking (Fornasier and Solombrino, 2014), finance (Carmona and Laurière, 2023), opinion dynamics (Liang and Wang, 2019), and artificial collective behavior (Gu et al., 2021; Cui et al., 2024), among others. In contrast with optimal control, mean field approximations have not

been used for OS problems, except for (Talbi et al., 2024; 2023) in the continuous time and continuous space setting, and computational methods have not yet been developed.

Our work makes a first step in this direction: focusing on discrete time, finite space MFOS models, we provide a theoretical foundation and then introduce two deep learning methods, which can solve MFOS with many states by learning optimal stopping decisions that are functions of the whole population distribution. We call these methods direct approach (DA) and dynamic programming approach (DP), and test them on several environments.

> **Main Contributions**   Our main contributions are twofold:
>
> *Theoretically*: (1) we prove that MFOS in discrete space and time yields an **approximate optimal stopping decision** for $N$-agent MAOS with a rate of $\mathcal{O}(1/\sqrt{N})$ (Thm 3.2); (2) we prove a **DPP** for MFOS by interpreting the model as a special kind of MFC problem (Thm 4.1).
>
> *Computationally*: (1) we propose **two deep learning methods** to solve MFOS problems, by learning the optimal stopping decision as a function of the whole population distribution (Alg. 1 and 2); (2) we illustrate the performance of both algorithms on **six environments** of increasing complexity, with distributions' dimension and time horizon up to 300 and 50 respectively.

To the best of our knowledge, this is the first work to study discrete-time, finite-space MFOS problems. Our theoretical results relay on the interpretation of MFOS problems as MFC problems, which provides a new perspective and opens up new direction to study MFOS problems. Additionally, it is the first time that computational methods are proposed to solve MFOS. This is a first step towards solving complex multi-agent optimal stopping problems with very large number of agents.

**Related works**   MFOS has been recently studied in continuous time and space from a purely theoretical view by Talbi et al. (2023; 2024) who studied the connection with finite-agent MAOS problems and characterized the solution of (continuous) MFOS using a PDE on the infinite-dimensional space of probability measures, which is intractable. Instead, we focus on *discrete time* scenarios with *finite state space* (i.e., an individual agent's state can take only finitely many different values), and hence the distribution is finite dimensional. This setting can be viewed as an approximation of the continuous setting. Deep learning methods have been proposed for discrete time *single-agent* OS problems. Becker et al. (2019) proposed to learn the stopping decision at each time using a deep neural network. Herrera et al. (2023) extended the approach using randomized neural networks. Damera Venkata and Bhattacharyya (2024) proposed to use recurrent neural networks to solve non-Markovian OS problems. Other approaches have been proposed, particularly for continuous time OS problems, such as learning the stopping boundary (Reppen et al., 2022). These single-agent OS approaches cannot be easily adapted to solve MAOS problems: The solution which consists in treating the whole system as one agent would lead to stopping all the agents at the same time, and single-agent methods do not capture the interdependence between agents. Furthermore, these approaches are not suitable to tackle continuous space MFOS problems as introduced by Talbi et al. (2023) because the value function must be a function of the population distribution, which leads to an infinite dimensional problem. For this reason, there are *no existing computational methods for MFOS*. In this work, we focus on a finite space setting and propose the first computational methods for MFOS problems by leveraging the aforementioned deep learning literature to tackle the (finite but) high dimensionality of the population distribution. Another difference with (Talbi et al., 2023; 2024) is that these work purely rely on an OS viewpoint, while we unveil a connection with MFC problems. This is a conceptual contribution of our work. Recently, MFC problems in discrete time and finite space have been studied using reinforcement learning methods (Gu et al., 2023; Motte and Pham, 2022; Carmona et al., 2023). However, here we focus on situations in which the model is known and we develop deep learning algorithms. This allows us to solve problems in much higher dimension (up to 300 for the neural network's input) than these works.

## 2   MODEL

When the number of agents tends to infinity an aggregation effect takes place, enabling us to represent the influence of the community using an "average" term, commonly referred to as the *mean field* term. As the number of agents approaches infinity, they become independent and identically

distributed (i.i.d.), and the behavior of each individual agent is determined by a stochastic differential equation (SDE) of McKean-Vlasov type. This phenomenon is often known as the "propagation of chaos" (Sznitman, 1991). The objective is to discern the properties of the solutions to the limiting problem. By integrating these properties into the formulation of the $N$-agent control framework, we can derive approximate solutions to the latter problem (for more theoretical background on MFC, see (Bensoussan et al., 2013; Carmona et al., 2013; Carmona and Delarue, 2018)).

## 2.1 MOTIVATION: FINITE AGENT MODEL

The mean field problem that we will solve is motivated by the $N$-agent problem that we are about to describe. Let $\mathcal{X}$ be a finite state space. Let us denote by $\mathcal{P}(\mathcal{X})$ the set of probability distributions on $\mathcal{X}$, and let $E$ be the set of realizations of the random noise. Let $T$ be a time horizon and let $N$ be the number of agents that are interacting.. Each agent $i$ has a state denoted by $X_n^i$ at time $n$. At time $n$, each agent stops with probability $p_n^i(\boldsymbol{X}_n^{\boldsymbol{\alpha}})$. We introduce $\alpha_n^i$ a random variable taking value 0 if the agent continues and 1 if it stops. We denote by $\pi_n^i(\cdot|\boldsymbol{X}_n^{\boldsymbol{\alpha}}) = Be(p_n^i(\boldsymbol{X}_n^{\boldsymbol{\alpha}}))$ its distribution, which is a Bernoulli distribution. We denote by $\boldsymbol{X}_n^{\alpha} = (X_n^1, \dots, X_n^N)$ and $\boldsymbol{\alpha} = (\alpha^1, \dots, \alpha^N)$ the vectors of states and stopping decisions at time $n$.

**Dynamics.** We assume that the agents are indistinguishable and interact in a symmetric fashion, i.e. through their empirical distribution $\mu_n^{N,\boldsymbol{\alpha}}(x) := \frac{1}{N}\sum_{i=1}^N \delta_{X_n^{i,\alpha}}(x)$, which is the proportion of agents at $x$ at time $n$ with $\delta$ the indicator function. The system evolves according to a transition function $F : \mathbb{N} \times \mathcal{X} \times \mathcal{P}(\mathcal{X}) \times E \to \mathcal{X}$. In particular: for every $i = 1, \dots, N$,

$$\begin{cases} X_0^{i,\boldsymbol{\alpha}} \sim \mu_0 \\ \alpha_n^i \sim \pi_n^i(\cdot|\boldsymbol{X}_n^{\boldsymbol{\alpha}}), \qquad X_{n+1}^{i,\boldsymbol{\alpha}} = \begin{cases} F(n, X_n^{i,\boldsymbol{\alpha}}, \mu_n^{N,\boldsymbol{\alpha}}, \epsilon_{n+1}^i), & \text{if } \sum_{m=0}^n \alpha_m^i = 0 \\ X_n^{i,\boldsymbol{\alpha}}, & \text{otherwise,} \end{cases} \end{cases} \quad (1)$$

where $\epsilon_n^i$ is a random noise impacting the evolution of agent $i$ and $m_0$ is the initial distribution.

Let us define the stopping time for agent $i$: $\tau^i = \inf\{n \geq 0 : \sum_{m=0}^n \alpha_m^i \geq 1\}$, which is the first time for player $i$ at which the decision is to stop.

**Objective function.** Let us consider a function $\Phi : \mathcal{X} \times \mathcal{P}(\mathcal{X}) \to \mathbb{R}$. $\Phi(x, \mu)$ denotes the cost that an agent incurs if she stops at $x$ and the current population distribution is $\mu$. The goal for all the $N$ agents is to collectively minimize the following social cost function:

$$J^N(\alpha^1, \dots, \alpha^N) = \mathbb{E}\left[\frac{1}{N}\sum_{i=1}^N \Phi(X_{\tau^i}^{i,\boldsymbol{\alpha}}, \mu_{\tau^i}^{N,\boldsymbol{\alpha}})\right]. \quad (2)$$

The problem consists in finding $(\alpha^1, \dots, \alpha^N) \in \arg\min J^N$. Next, we give an example.

**Motivating Example:** We take state space $\mathcal{X} = \{1, 2, 3, 4, 5, 6, 7\}$ with boundaries (i.e., in 1 agents cannot move left and in 7 they cannot move right), time horizon $T = 3$, transition function $F(n, x, \mu, \epsilon) = x + \epsilon$, where $\epsilon = 0$ with probability $p = 1/2$, $\epsilon = 1$ with probability $p = 1/4$ and $\epsilon = -1$ with probability $p = 1/4$. Following (1), the dynamics of agent $i$ is: $X_{n+1}^i = X_n^i + \epsilon_{n+1}^i$ if the agent does not stop, and $X_{n+1}^i = X_n^i$ otherwise. All agents start in $x = 4$. We define a target distribution $\rho_{\text{target}} = \frac{1}{2}\delta_4 + \frac{1}{4}\delta_5 + \frac{1}{4}\delta_3$. If the agent $i$ stops at time $n$, then she is incurred the cost: $\Phi(X_n^i, \mu_n^N) = \sum_{x \in \mathcal{X}} |\mu_n^N(x) - \rho_{\text{target}}(x)|^2$, which is smaller if the agent stops when the population distribution matches the target one. Notice that some agents might have to stop even though the target distribution is not matched, so that other agents can later have a lower cost because this is a *cooperative* task. Solving exactly this problem (i.e., finding the optimal stopping time for every agent) is very complex. Our approach consists in considering the mean field problem, which leads to an efficient approximate solution (see Example 4 in Section 6).

> **Challenges:** Single-agent methods cannot be readily applied to the multi-agent setting since they cannot capture the interdependence due to the distribution in the cost and in the dynamics. In particular, in the multi-agent setting, we allow agents to stop at different times. When the number of agents is very large, computing exactly the optimal stopping times is infeasible.

> Mean field optimal stopping (MFOS) can intuitively provide an approximate solution but (1) this needs to be justified and (2) scalable numerical methods for MFOS need to be developed.

## 2.2 MEAN FIELD MODEL

As mentioned earlier, if we let the number of players tend to infinity, we expect, thanks to propagation of chaos type results, that the states will become independent and each state will have the same evolution, which will be a non-linear Markov chain. More precisely, passing formally to the limit in the dynamics (1), we obtain the following evolution:

$$
\begin{cases}
X_0^\alpha \sim \mu_0 \\
\alpha_n \sim \pi_n(\cdot|X_n^\alpha) = Be(p_n(X_n^\alpha)), \ X_{n+1}^\alpha = \begin{cases} F(n, X_n^\alpha, \mu_n^\alpha, \epsilon_{n+1}), & \text{if } \sum_{m=0}^n \alpha_m = 0 \\ X_n^\alpha, & \text{otherwise,} \end{cases}
\end{cases} \tag{3}
$$

where $p_n(x)$ denotes the probability with which the agent continues if she is in state $x$ at time $n$, and $\mu_n^\alpha$ is the distribution of $X_n^\alpha$ itself, which we may also denote by $\mathcal{L}(X_n^\alpha)$.

We want to emphasize the fact that the introduction of randomized stopping times for individual agents is crucial for our purpose; see the example in Appx. A.1.

We can define, in the same way we did before, the first time at which the control $\alpha$ is 1 as $\tau := \inf\{n \geq 0 : \sum_{m=0}^n \alpha_m \geq 1\}$. Then the social cost function in the mean field problem is defined as:

$$
J(\alpha) = \mathbb{E}\left[\Phi(X_\tau^\alpha, \mathcal{L}(X_\tau^\alpha))\right]. \tag{4}
$$

Notice that here the expectation has the effect of averaging over the whole population, so there is no counterpart to the empirical average that appears in the finite agent cost (2). To stress the dependence on the initial distribution, we will sometimes write $J(\alpha, m_0)$.

## 2.3 MEAN FIELD MODEL WITH EXTENDED STATE

A key step towards building efficient algorithms is dynamic programming, which relies on Markovian property. However, in its current form the above problem is not Markovian. This makes the problem *time-inconsistent*. To make the system Markovian, we need keep track of the information about whether the player's process has been stopped in the past. This information is not contained in the state so we need to extend the state. Let $A^\alpha = (A_n^\alpha)_{n=0,\dots,T}$ the process such that $A_n^\alpha = 0$ if the agent has *already* stopped before time $n$, and 1 otherwise. We can interpret this process as the "Alive" process, while $\alpha$ stands for the "action", namely, to stop or not. So $A_n^\alpha = 1$ means the agent has not stopped yet; when the agent stops, $\alpha_n = 1$ and $A_{n+1}^\alpha$ switches to 0. It is important to notice that if the agent is stopped precisely at time $n$ then, we still have $A_n^\alpha = 1$ but $A_m^\alpha = 0$ for every $m > n$. We define the *extended state* as: $Y_n^\alpha = (X_n^\alpha, A_n^\alpha)$, which takes value in the extended state space $\mathcal{S} := \mathcal{X} \times \{0, 1\}$. Then, the dynamics (3) of the representative player can rewritten as:

$$
\begin{cases}
X_0^\alpha \sim \mu_0, \qquad A_0^\alpha = 1 \\
\alpha_n \sim \pi_n(\cdot|X_n^\alpha) = Be(p_n(X_n^\alpha)) \\
A_{n+1}^\alpha = A_n^\alpha \cdot (1 - \alpha_n) \\
X_{n+1}^\alpha = \begin{cases} F(n, X_n^\alpha, \mathcal{L}(X_n^\alpha), \epsilon_{n+1}), & \text{if } A_n^\alpha \cdot (1 - \alpha_n) = 1 \\ X_n^\alpha, & \text{otherwise.} \end{cases}
\end{cases} \tag{5}
$$

The idea of extending the state using the extra information is similar to Talbi et al. (2023) in continuous time and space. The mean field social cost (4) can rewritten as:

$$
J(\alpha) = \mathbb{E}\left[\sum_{m=0}^T \Phi(X_m^\alpha, \mathcal{L}(X_m^\alpha)) A_m^\alpha \alpha_m\right] \tag{6}
$$

Actually, notice that the expectation amounts to taking a sum with respect to the extended state's distribution. Let us denote by $\nu_n^p = \mathcal{L}(Y_n^\alpha)$ the distribution at time $n$. We are going to denote $\nu_X^p$ the first marginal of $\nu^p$ (sometimes also denoted by $\mu$). Note that it does not really depend on $\alpha$ but

only on the stopping probability $p$, so we use the superscript $p$ when referring to $\nu$. This distribution evolves according to the mean field dynamics:

$$\begin{cases} \nu_0^p(x,0) = 0, \quad \nu_0^p(x,1) = \mu_0(x), \qquad x \in \mathcal{X}, \\ \nu_{n+1}^p = \bar{F}(\nu_n^p, p_n), \end{cases} \tag{7}$$

where the function $\bar{F}$ is defined as follows. We denote by $\mathcal{H}$ the set of all function $h : \mathcal{X} \to [0,1]$, which represents a stopping probability (for each state). Then, $\bar{F} : \{0, \dots, T\} \times \mathcal{P}(\mathcal{S}) \times \mathcal{H} \to \mathcal{P}(\mathcal{S})$ is defined by: for every $x \in \mathcal{X}, a \in \{0,1\}$, $\bar{F}(\nu, h)$ is the distribution generated by doing one step, starting from $\nu$ and using the stopping probabilities $h$. Mathematically,

$$(\bar{F}(\nu,h))(x,a) = \left( \nu(x,0) + \nu(x,1)h(x) \right)(1-a) + \left( \sum_{z \in \mathcal{X}} \nu(z,1) \left( q_{z,x}^\nu(1-h(z)) \right) \right)a, \tag{8}$$

where $\mathcal{Q}^\nu = (q_{z,x}^\nu)_{z,x \in \mathcal{X}}$ is the transition matrix associated to the unstopped process $X$, i.e. $q_{z,x}^\nu$ is the probability to go from the state $z$ to the state $x$ knowing that we are not going to stop in $x$. Notice that in general the transitions may depend on $\nu$ itself. So the last equation can be written more succinctly in a matrix-vector product but the transition matrix depends on $\nu$ itself, which is why this type of dynamics is sometimes referred to a *non-linear* dynamics. The mean field social cost can be rewritten purely in terms of the distribution as follows:

$$J(p) = \sum_{m=0}^T \sum_{(x,a) \in \mathcal{S}} \nu_m^p(x,a) \Phi(x, \mu_m^p) a p_m(x), \tag{9}$$

where $p : \{0, \dots, T\} \times \mathcal{X} \to [0,1]$ is the function that associates to every time step and state the probability to stop (in that state at that time). Let us define $\mathcal{P}_{0,T}$ the set of all such functions.

The link with the above formulation is that $\alpha_n(x)$ is distributed according to $Be(p_m(x))$, and $\nu_m^p := \mathcal{L}(Y_m^\alpha)$ is the extended state's distribution. Moreover, $\nu_m^p(x,0)$ is the mass in $x$ that has stopped. Last, $\mathcal{L}(X_m^\alpha) = \mu_m^\alpha(x) = \sum_{a \in \{0,1\}} \nu_m^p(x,a)$ is the first marginal of this distribution.

# 3 APPROXIMATE OPTIMALITY FOR FINITE-AGENT MODEL

In this section, we aim to address the following question: *"Is the mean-field model capable of solving the original problem of $N$ agents, at least approximately?"*. Specifically, we demonstrate that the MFOS solution provides an approximately optimal solution for the finite-agent MAOS problem. The main assumption we use is:

**Assumption 3.1.** Let $L_p > 0$ and let us define $\mathcal{P} := \{p : \{0, \dots, T\} \times \mathcal{X} \times \mathcal{P}(\mathcal{S}) \to [0,1] : p$ is $L_p$-Lipschitz$\}$, the set of all possible admissible policies $p$. Assume that the *mean field* dynamics $\bar{F}$ described in (8) is $L_{\bar{F}}$ - Lipschitz. Assume also that the function $\Psi : \mathcal{P}(\mathcal{X} \times \{0,1\}) \times \mathcal{P}(\mathcal{X}) \to \mathbb{R}$ defined as $\Psi(\nu, h) := \sum_{(x,a) \in \mathcal{S}} \nu(x,a) \Phi(x, \nu_X) a h(x)$ is $L_\Psi$-Lipschitz.

Assuming Lipschitz dynamics, cost and policies is classical in the literature on mean field control problems, see e.g. (Mondal et al., 2022; Pásztor et al., 2023; Cui et al., 2023) and can be achieved using neural networks (Araujo et al., 2022).

Due to space constraints, we simply provide an informal statement here. The precise statement is deferred to Appx. A.2, see Theorem A.3, along with the detailed setting and notations.

**Theorem 3.2** ($\varepsilon$-approximation of the $N$-agent problem). *Suppose Assumption 3.1 holds. If $p^*$ is the optimal policy for the MFOS problem and $\hat{p}$ is the optimal policy for the $N$-agent problem (when all the agents use the same policy), then: as $N \to +\infty$, $J^N(p^*, \dots, p^*) - J^N(\hat{p}, \dots, \hat{p}) \to 0$, with rate of convergence $\mathcal{O}\left(1/\sqrt{N}\right)$ (the explicit bound is in the proof).*

A key step in the proof consists in analyzing the difference between the $N$-agent dynamics and the mean-field dynamics under a stopping policy, see Lemma A.1 ("Convergence of the measure") in appendix.

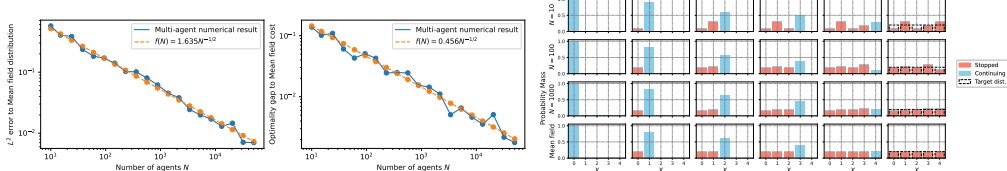

Figure 1: MFOS v.s. MAOS. We use the stopping probability function learned by Algorithm 1 for MFOS to simulate the multi-agent OS.

Theorem 3.2 is further supported through empirical evidence as is shown in Fig. 1, where we apply the stopping probability function learned by Algorithm 1 on MFOS in Example 1 (see Section 5 and 6 for details) to the $N$-agent problem with varying $N$ (see Appx. E.1). We compute the $L^2$ distance of multi-agent empirical distribution to mean-field distribution and the optimality gap between multi-agent and mean-field cost, both averaged over 10 runs. The plots demonstrate a clear decay rate of order $N^{-1/2}$. This theorem justifies that MFOS is not only an intrinsically interesting problem, but the solution to MFOS also serves as an approximate solution to the corresponding MAOS problem. In the sequel, we will focus on solving the MFOS problem.

## 4 DYNAMIC PROGRAMMING

Our motivation for developing a dynamic programming principle (DPP) for our formulation comes from both the literature and numerical purposes. Dynamic programming (DP) appears very often in the literature, encompassing fields such as economics, finance, development of computer programs to the ability of a computer to master the game of chess, Go, and many others. In the control theory of a dynamic system in particular, it has been studied and used very often to find solutions to a given optimization problem. Moreover, implementing an algorithm that founds on DPP often leads to precise optimal solutions that perform better than other methods.

We introduce the dynamical form of the social cost (9) as:

$$V_n(\nu) := \inf_{p \in \mathcal{P}_{n,T}} J(p(x), \nu) := \inf_{p \in \mathcal{P}_{n,T}} \sum_{m=n}^{T} \sum_{(x,a) \in \mathcal{S}} \nu_m^{p,\nu,n}(x,a) \Phi(x, \mu_m^{p,\nu,n}) a p_m(x), \qquad (10)$$

where $\mathcal{P}_{n,T}$ is the set of all possible function $p : \{n, \dots, T\} \times \mathcal{X} \to [0,1]$ and $\nu^{p,\nu,n}$ denotes the distribution of the process that starts at time $n$ with a given distribution $\nu$; it satisfies (7) but starting at time $n$ instead of 0 with $\nu_n^{p,\nu,n} = \nu$. The optimal value at time 0 will be denoted: $V^*(\nu) = V_0(\nu)$, which is also equal to $\inf_p J(p, \nu)$. We can now state and prove the following DPP.

**Theorem 4.1** (Dynamic Programming Principle). *For the dynamics given by (5) and the value function given by (10) the following dynamic programming principle holds:*

$$\begin{cases} V_T(\nu) = \sum_{(x,a) \in \mathcal{S}} \nu(x,a) \Phi(x, \nu_X) a, \\ V_n(\nu) = \inf_{h \in \mathcal{H}} \sum_{(x,a) \in \mathcal{S}} \nu(x,a) \Phi(x, \nu_X) a h(x) + V_{n+1}(\bar{F}(\nu, h)), \qquad n < T, \end{cases} \qquad (11)$$

*where $\nu_X$ is the first marginal of the distribution $\nu$, i.e., $\nu_X(x) = \nu(x,0) + \nu(x,1)$. The sequence of optimizers define an optimal stopping decision that we will denote by $h^* : \{0, \dots, T-1\} \times \mathcal{X} \times \mathcal{P}(\mathcal{S}) \to [0,1]$ and satisfies: for every $n \in \{0, \dots, T-1\}$ and every $\nu \in \mathcal{P}(\mathcal{S})$, $V_n(\nu) = \sum_{(x,a) \in \mathcal{S}} \nu(x,a) \Phi(x, \nu_X) a h_n^*(x, \nu) + V_{n+1}(\bar{F}(\nu, h_n^*(x, \nu)))$.*

To prove this result, we will show that we can reduce the problem to a mean field optimal control problem in discrete time and continuous space. See details in Appx. B. Dynamic programming for MFC problem (Laurière and Pironneau, 2014; Pham and Wei, 2017) and mean field MDPs (Gu et al., 2023; Motte and Pham, 2022; Carmona et al., 2023) have been extensively studied, and DPP for continuous time MFOS has been established by Talbi et al. (2023) using a PDE approach. However, to the best of our knowledge, this is the first DPP result for MFOS problems in discrete time. It serves as a building block for one of the deep learning methods proposed below.

Actually we can show that this DPP still holds for a restricted class of randomized stopping times in which all the agents (regardless of their own state) have the same probability of stopping. Let $\tilde{\mathcal{P}}_{n,T}$ be the set of $p : \{0, \ldots, T\} \to [0, 1]$. Notice that here $p_n$ does not depend on the individual state $x$. At every time step $n = m$ every agent has the same probability to stop $p_m$, i.e for every $x \in \mathcal{X}$ at time $n = m$, $p_n(x) = p_n$. We call this set as *synchronous* stopping times. Let us define the value:

$$\tilde{V}_n(\nu) := \inf_{p \in \tilde{\mathcal{P}}_{n,T}} J(p, \nu) := \inf_{p \in \tilde{\mathcal{P}}_{n,T}} \sum_{m=n}^{T} p_m \sum_{(x,a) \in \mathcal{S}} \nu_m^{p,\nu,n}(x,a) \Phi(x, \mu_m^{p,\nu,n}) a.$$

**Theorem 4.2.** *For the setting of synchronous stopping times, the value function satisfies:*

$$\begin{cases} \tilde{V}_T(\nu) = \sum_{(x,a) \in \mathcal{S}} \nu(x,a) \Phi(x, \nu_X) a, \\ \tilde{V}_n(\nu) = \inf_{h \in [0,1]} \sum_{(x,a) \in \mathcal{S}} \nu(x,a) \Phi(x, \nu_X) ah + V_{n+1}(\bar{F}(\nu, h)), \qquad n < T. \end{cases} \tag{12}$$

The proof follows the same argument as the one of Theorem 4.1 so we omit it.

## 5 Algorithms

To address the MFOS problem numerically, we propose two approaches based on two different formulations. As the most naive approach, we can attempt to directly minimize the mean-field social cost $J(p)$ stated in (9), where we optimize over all the possible stopping probability functions $p : \{0, \ldots, T\} \times \mathcal{X} \to [0, 1]$. A more ideal treatment is to leverage the Dynamic Programming Principle (DPP) discussed in Theorem 4.1 and solve for the optimal stopping probability using induction backward in time. For each of the timestep $n$, we implicitly learn the true value function $V_n(\nu)$ by solving the optimization problem in (11), where we search over all possible one-step stopping probability function $h : \mathcal{X} \to [0, 1]$ for each time $n$. We refer to the method of directly optimizing mean-field social cost as the direct approach (DA) and the attempt to solve MFOS via backward induction of the DPP approach. Short versions of the pseudocodes are presented in Alg. 1 and 2. Long versions are in Appx. C (see Alg. 3 and 4). To alleviate the notations, we denote: $\bar{\Phi}(\nu, h) = \sum_{x \in \mathcal{X}} \nu(x, 1) \Phi(x, \nu_X) h(x)$, which represents the one-step mean field cost. In the code, optim_up denotes one update performed by the optimizer (e.g. Adam in our simulations).

---

**Algorithm 1** Direct Approach (DA)

**Require:** time-dependent stopping decision neural network: $\psi_\theta : \{0, \ldots, T\} \times \mathcal{X} \times \mathcal{P}(\mathcal{S}) \to [0, 1]$, max number of training iteration $N_{\texttt{iter}}$
1: **for** $k = 0, \ldots, N_{\texttt{iter}} - 1$ **do**
2:     Sample initial $\nu_0^p$
3:     **for** $n = 0, \ldots, T$ **do**
4:        $p_n(x) = \psi_\theta(x, \nu_n^p, n; \theta_k), x \in \mathcal{X}$
5:        $\ell_n = \bar{\Phi}(\nu_n^p, p_n)$
6:        $\nu_{n+1}^p = \bar{F}(\nu_n^p, p_n)$
7:     $\ell = \sum_{n=0}^{T} \ell_n$
8:     $\theta_{k+1} = \texttt{optim\_up}(\theta_k, \ell(\theta_k))$
9: Set $\theta^* = \theta_{N_{\texttt{iter}}}$
10: **return** $\psi_{\theta^*}$

---

**Algorithm 2** Dynamic Programming (DP)

**Require:** stopping decision neural networks: $\psi_\theta^n : \mathcal{X} \times \mathcal{P}(\mathcal{S}) \to [0, 1]$ for $n \in \{0, \ldots, T - 1\}$, max training iteration $N_{\texttt{iter}}$.
1: Set $\psi_\theta^T = 1$
2: **for** $n = T - 1, \ldots, 0$ **do**
3:     **for** $k = 0, \ldots, N_{\texttt{iter}} - 1$ **do**
4:        Sample $\nu_n^p$
5:        **for** $m = n, \ldots, T$ **do**
6:           **if** $m = n$ **then**
7:              $p_m(x) = \psi_\theta^m(x, \nu_m^p; \theta_k^n)$
8:           **else**
9:              $p_m(x) = \psi_\theta^m(x, \nu_m^p; \theta^{m,*})$
10:           $\ell_m = \sum_x \nu_m^p(x, 1) \Phi(x, \mu_m) p_m(x)$
11:           $\nu_{m+1}^p = \bar{F}(\nu_m^p, p_m)$
12:        $\ell = \sum_{m=n}^{T} \ell_m$
13:        $\theta_{k+1}^n = \texttt{optim\_up}(\theta_k^m, \ell(\theta_k^n))$
14:     Set $\theta^{n,*} = \theta_{N_{\texttt{iter}}}^n$
15: **return** $(\psi_{\theta^{n,*}}^n)_{n=0, \ldots, T}$

---

## 6 Experiments

In this section, we present 6 experiments of increasing complexity to validate our proposed method and demonstrate its potential applications. Due to space constraints, two of them have been included in Appx. E. It is important to emphasize that each experiment reflects a distinct scenario, varying both in dynamics (random, deterministic; with or without mean-field interactions) and in the cost

function (with or without mean-field dependence). This provides a comprehensive overview of the method's versatility and potential applications. Eventually, we present a task of spatial dimension 300 (i.e., neural network's input dimension) and time horiwon 50 with a random obstacle dynamics, motivated by applications to a fleet of drones which have to match a target distribution. We solve all 6 environments with both algorithms (the details are in Appx. E).

**Problem Dimensions:** For the problem dimension, we count it as the sum of the dimension of the information input to the neural network. Since the state is in $\mathcal{X}$, which is finite, we encode it as a one-hot vector in $\mathbb{R}^{|\mathcal{X}|}$ before passing it to the neural network to ensure differentiability. For the mean-field distribution with stopped and non-stopped parts, it is an element of the $(2|\mathcal{X}| - 1)$-simplex, and is represented as a non-negative vector in $\mathbb{R}^{2|\mathcal{X}|}$. Therefore, MFOS tasks are intrinsically of spatial dimension $|\mathcal{X}| + 2|\mathcal{X}| = 3|\mathcal{X}|$, where $|\mathcal{X}|$ is the dimension of an individual agent's state space.

**Comparison of the Two Proposed Algorithms:** While in theory both algorithms are equally capable of tackling MFOS problems, in practice these algorithms have advantages in different settings. Empirically, we found that the optimal stopping decision is learned faster by DA than by DP, when compute power is not a restriction. However, DA requires differentiating through the whole trajectory at each gradient step, which requires a large amount of memory when the dimension is high. The minimum required memory for training with DA increases with $T$, whereas DP requires only constant order memory that is independent of the time horizon, since it trains time step per time step. Therefore, when targeting an MFOS problem with a long time horizon $T$, DPP becomes more efficient, at least memory-wise. Similar observations were made in the context of continuous time optimal control by Germain et al. (2022).

**Example 1** (Towards the uniform) and **Example 2** (Rolling a die), on a 1D gridworld state space, are described in details in Appx. E.1 and Appx. E.2 respectively due to space constraint.

**Example 3: Crowd Motion with Congestion.** This example extends the setting of Example 2 by incorporating a congestion term into the dynamics. The outcome of the die takes the role of the noise $\epsilon \sim \mathcal{U}(\mathcal{X})$ where $\mathcal{X} = \{1, 2, 3, 4, 5, 6\}$. The system starts in the initial distribution $\eta = \frac{1}{4}\delta_1 + \frac{1}{4}\delta_2 + \frac{1}{2}\delta_5$, and evolves according to the dynamics (5) with $\mu_0 = \eta$, and $F(n, x, \mu, \epsilon) = \epsilon$ where we are going to introduce a term of *congestion* multiplying the probability of moving by $(1 - C_{\text{cong}}\mu(x))$ to model the fact that it is difficult to move from a state $x$ if the distribution is concentrated in that state (details in Appx. E.3) . The social cost function associated to this scenario is $\Phi(x, \mu) = x$. Time horizon is set to $T = 4$. We executed the experiment without congestion (see Appx. E.2) and we expect congestion to slow down the movement. DR results are shown in Fig. 2.

This example demonstrates that two classes of stopping times (synchronous and asynchronous) can lead to very different optimal stopping decisions and induce distributions. Although the true value is unknown, the results indicate that synchronous stopping times yield a higher value, while asynchronous stopping times lead to a significant reduction in the cost. Additionally, in the asynchronous case, congestion leads to reduced movement, as observed between time 0 and time 1 in state 4. See Appx. E.3 for the DPP results.

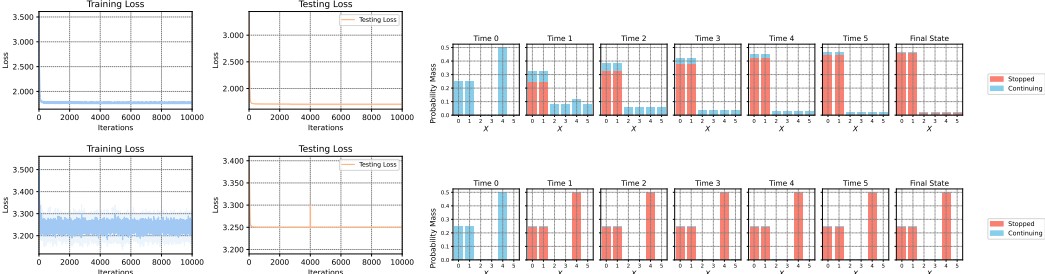

Figure 2: Example 3. DA results, asynchronous vs synchronous stopping times. Left: comparison of training and testing losses for asynchronous stopping times (top) and synchronous stopping times (bottom). Right: Comparison of the evolution of the distribution after training (asynchronous stopping class on top, synchronous stopping class on bottom).

**Example 4: Distributional Cost.** This example extends, at the mean field level, the motivating example described at the end of Section 2.1. Based on Theorem 3.2, the mean field solution provides a good approximation of the $N$-agent problem. DR and DPP results are shown in Fig. 3 and 4 respectively. The results for synchronous stopping time are shown in Appx. E.4.

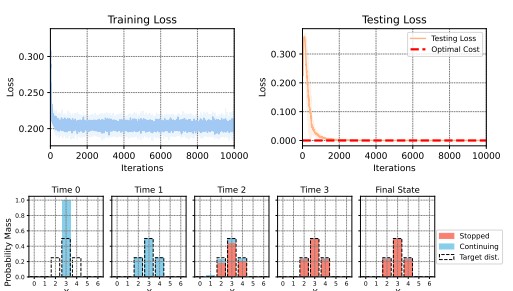

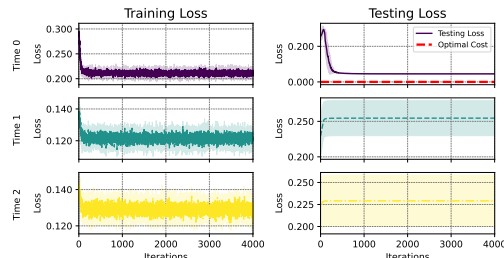

Figure 3: Example 4. DA results, asynchronous stopping. Top: training and testing losses. Bottom: evolution of the distribution after training.

Figure 4: Example 4. DPP results, asynchronous stopping. Training and testing losses.

**Example 5: Towards Uniform in Dimension 2.** This example extends Example 1 (see Appx. E.1) to two dimensions, demonstrating how the algorithm performs in higher-dimensional settings. We take state space $\mathcal{X} = \{0, 1, 2, 3, 4\} \times \{0, 1, 2, 3, 4\}$, time horizon $T = 4$, transition function $F(n, x, \mu, \epsilon) = x + (1, 0)$ which means that the agent deterministically moves to the state on the right on the same row, with boundary at $x = 4$, and cost function $\Phi(x, \mu) = \mu(x)$ which depends on the mean field only through the state of the agent (this is sometimes called local dependence). For the testing distribution, we take a distribution concentrated on state $x = 0$, denoted as $\mu_0 = \delta_0$. Fig. 5 shows that the distribution evolves towards a uniform distribution across each row, as expected, and also illustrates the optimal strategy (decision probability) required to achieve this outcome. Results for the DPP algorithm and the synchronous stopping are in Appx. E.5.

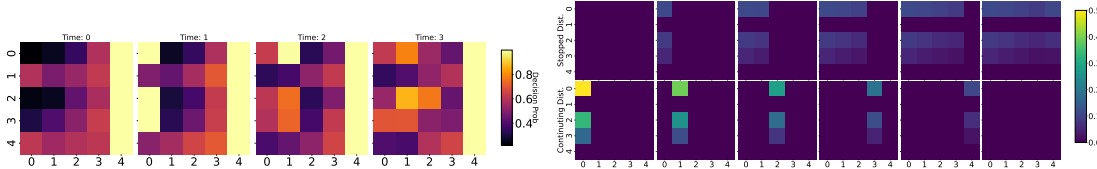

Figure 5: Example 5. DA results, asynchronous stopping. Left: stopping decision probability. Right: evolution of the distribution after training.

**Example 6: Matching a Target with a Fleet of Drones.** We conclude with a more realistic and complex example to showcase the potential applications of our algorithms. This example aims to align a fleet of drones with a given target distribution at terminal time $T$, starting from a random initial distribution. To make this experiment more interesting, we expand the framework described so far by considering a different type of cost and by including a noisy obstacle hindering the drones' movements (see Appx. E.6 for the mathematical formulation). We take $\mathcal{X} = \{0, \ldots, 9\} \times \{0, \ldots, 9\}$ that represents a $10 \times 10$ grid. Hence, the neural network's input is of dimension $3|\mathcal{X}| = 300$. The system follows the dynamics that diffuse uniformly over the possible neighbors, where the possible neighbors of $x \in \mathcal{X}$ are defined as $x \pm (0, 1)$ or $x \pm (1, 0)$ if the resulting state is still an element of $\mathcal{X}$. Moreover, we introduce extra stochasticity into the dynamics by placing an obstacle at a random state on the grid at each time step. The location is uniformly selected from $\mathcal{X}$ and is viewed as a *common noise* affecting the dynamics of all the agents. This introduces additional complexity in the learning problem because even for a fixed stopping decision rule, the evolution of the population is stochastic. We consider the target distribution $\rho$ to be the uniform distribution over the grid of the letter "M", "F", "O", and "S" respectively, and we set the terminal cost $g_\rho(\nu) = \sum_{x \in \mathcal{X}} |\nu(x) - \rho(x)|^2$. We choose the time horizon $T = 50$. Fig. 6 shows that the learned stopping probability from Algorithm 2 successfully drives the initial distribution into the shape of the letter

"M". Another important aspect of the algorithms' outcome is that the learned stopping decisions are *agnostic to the initial distribution* in the sense that the same stopping decision rule can be used on different initial distributions and always leads to matching the target distribution. Fig. 7 shows the terminal distributions under random initial testing distribution: the learned stopping probability function is robust to any test distribution used at inference time. Results for the DA algorithm are shown in Appx. E.6.

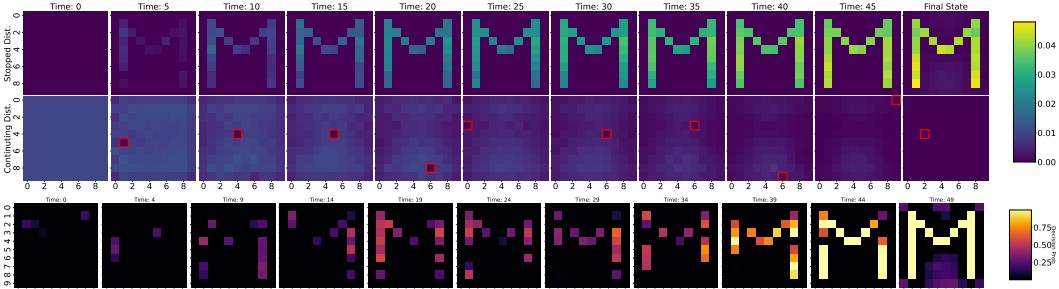

Figure 6: Example 6. DPP results, asynchronous stopping. Match the Letter "M" in $10\times10$ grid with common noise. We plot the stopped distribution, continuing distribution, and decision probability function every $5$ timestep. The marked red square indicates the random obstacles (common noise).

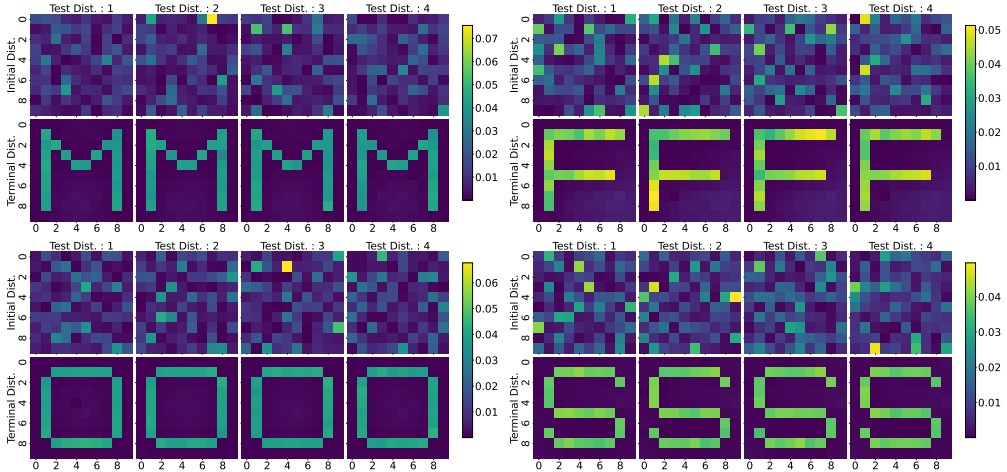

Figure 7: Example 6. DPP results, asynchronous stopping. Match the Letter "M", "F", "O", and "S". Tested with the randomly sampled initial distribution.

## 7 CONCLUSION

We proposed a discrete-time, finite state MAOS problem with randomized stopping times and its mean field version. We proved that the latter is a good approximation of the former, and we established a DPP for MFOS. These new problems cannot be tackled using traditional PDE approaches or adapting previous methods for single-agent OS problems. To overcome these challenges, we proposed two deep learning methods and evaluated their performance over six different scenarios. When an analytical solution is available, we demonstrated that our methods recover this solution in only a few iterations. In more complex environments, our approach is able to effectively solve the task with high performance. The approach presented in this work can be effectively extended to other contexts and applications, especially given the growing importance of MAOS problems.

**Limitations and Future Works:** First, we did not prove convergence of the algorithms due to the difficulty of analyzing deep networks. We also left for future work a detailed analysis of the comparison between synchronous and asynchronous stopping. Last, we would like to continue the numerical experimentation on more complex, real-world examples.

**Reproducibility Statement:** All the experimental details about computational resources and hyperparameters choices are provided in the appendix due to space limitation.

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

# A $N$-AGENT COOPERATIVE OPTIMAL STOPPING

## A.1 WHY DO WE NEED RANDOMIZATION IN THE CONTROL? AN EXAMPLE

We want to show with an example that the extension to randomized stopping times is necessary in the mean-field formulation, because when we try to plug an optimal strategy into the $N$-agent problem, we notice that the latter is no longer optimal.

*Example* 1 (Randomized is better). Let consider the following scenario: we take the state space $\mathcal{X} = \{T, C\}$ and initial distribution $\mu_0 = 3/4\delta_T + 1/4\delta_C$; transition function $F(T, x, \mu, \epsilon) = C$, $F(C, x, \mu, \epsilon) = T$, meaning that the system at any time step, can stop or switch the state. We take as social cost:

$$\Phi(x, \mu) = \begin{cases} 1 & \text{if } \mu(x) \leq 1/2 \\ 5 & \text{if } \mu(x) > 1/2. \end{cases} \tag{13}$$

Notice that without allowing the randomized stopping the value is $V^* = 3/4 \cdot 5 + 1/4 \cdot 1 = 4$, which corresponds to stop all the distribution ( in every state) at time $n = 0$. In the end, this formulation cannot reflect the optimum in the association of $N$ agents. Indeed when we plug this policy into the $N$ agent formulation we obtained the value $V^N = 1/N(3N/4 \cdot 5 + N/4 \cdot 1) = 4$, which is not optimal since we can use the strategy ( which is going to be optimal for the $N$-agent problem) to stop, at time 0, only the $1/3$ of players in state $T$, allowing the others to change state. This leads to a final configuration of $m_1 = 1/2\delta_T + 1/2\delta_C$ and a value of $V^{*,N} = 1/N(N/4 \cdot 5 + 3N/4 \cdot 1) = 2 < V^N = 4$.

In particular, we want to emphasize the fact that, without allowing a randomized stopping time in the MF formulation, we find an optimal state-dependent strategy, which corresponds , in the problem with finite agents, to the fact that every player in the same state will have the same stopping time.

## A.2 PROOF OF THEOREM 3.2

This section demonstrates that solving the optimal control problem at the asymptotic regime for the number of agents tending to infinity allows one to find the solution to the multi-agent problem by including the solution found at the regime in the latter. This is of fundamental importance in applications as it allows a simpler and clearer situation to be analyzed for the purpose of solving a complicated problem. Let us recall the $N$-agent formulation. We are going to work in the framework where the central planner use the same policy $p$ to control each agent. We suppose Assumption 3.1 holds.

Let us fix the following notation $\nu_m^{N,p} := \frac{1}{N} \sum_{i=1}^{N} \delta_{Y_m^{i,\alpha}}$ and $\nu_m^p := \mathcal{L}(Y_m^\alpha)$ .

$$\begin{cases} X_0^{i,\boldsymbol{\alpha}} \sim \mu_0, \qquad A_0^{i,\boldsymbol{\alpha}} = 1 \\ \alpha_n^i \sim \pi_n^i(\cdot | X_n^{i,\boldsymbol{\alpha}}) = Be(p_n(X_n^{i,\boldsymbol{\alpha}})) \\ A_{n+1}^{i,\boldsymbol{\alpha}} = A_n^{i,\boldsymbol{\alpha}} \cdot (1 - \alpha_n^i) \\ X_{n+1}^{i,\boldsymbol{\alpha}} = \begin{cases} F(n, X_n^{i,\boldsymbol{\alpha}}, \frac{1}{N}\sum_{j=0}^{N} \delta_{Y_n^{j,\alpha}}, \epsilon_{n+1}^i), & \text{if } A_n^{i,\boldsymbol{\alpha}} \cdot (1 - \alpha_n^i) = 1 \\ X_n^{i,\boldsymbol{\alpha}}, & \text{otherwise.} \end{cases} \end{cases} \tag{14}$$

The social cost is defined as:

$$J^N(p) := J^N(p, \ldots, p) := \frac{1}{N} \sum_{i=1}^{N} \mathbb{E}\left[\sum_{m=0}^{T} \Phi(X_m^{i,\boldsymbol{\alpha}}, \frac{1}{N}\sum_{i=0}^{N}\delta_{X_m^{i,\boldsymbol{\alpha}}})A_m^{i,\boldsymbol{\alpha}}\alpha_m^i\right] =$$

$$= \mathbb{E}\left[\sum_{m=0}^{T} \frac{1}{N}\sum_{i=1}^{N}\Phi(X_m^{i,\boldsymbol{\alpha}}, \frac{1}{N}\sum_{i=0}^{N}\delta_{X_m^{i,\boldsymbol{\alpha}}})A_m^{i,\boldsymbol{\alpha}}\alpha_m^i\right] =$$

$$= \mathbb{E}\left[\sum_{m=0}^{T} \sum_{(x,a)\in\mathcal{S}} \nu_m^{N,p}(x,a)\Phi(x,\nu_{X,m}^{N,p})a p_m(x)\right] = \tag{15}$$

$$= \mathbb{E}\left[\sum_{m=0}^{T} \Psi(\nu_m^{N,p}, p_m(\nu_m^{N,p}))\right]$$

The asymptotic problem is written as:

$$\begin{cases} X_0^\alpha \sim \mu_0, \qquad A_0^\alpha = 1 \\ \alpha_n \sim \pi_n(\cdot|X_n^\alpha) = Be(p_n(X_n^\alpha)) \\ A_{n+1}^\alpha = A_n^\alpha \cdot (1-\alpha_n) \\ X_{n+1}^\alpha = \begin{cases} F(n, X_n^\alpha, \mathcal{L}(X_n^\alpha), \epsilon_{n+1}), & \text{if } A_n^\alpha \cdot (1-\alpha_n) = 1 \\ X_n^\alpha, & \text{otherwise,} \end{cases} \end{cases} \tag{16}$$

where the social cost is defined as:

$$J(p) := \sum_{m=n}^{T} \sum_{(x,a)\in\mathcal{S}} \nu_m^{p,\nu,n}(x,a)\Phi(x,\nu_{X,m}^p)a p_m(x) =$$

$$= \sum_{m=n}^{T} \Psi(\nu_m^p, p_m(\nu_m^p)). \tag{17}$$

Let us recall that $\mathcal{P} := \{p : \{0, \ldots, T\} \times \mathcal{X} \times \mathcal{P}(\mathcal{S}) \to [0,1] : p \text{ is } L_p\text{-Lipschitz}\}$, the set of all possible admissible policies $p$. From now we are going to use the notation $\|\cdot\|$ for the norm associated to the total variation distance. Firstly we want to prove the at time time $n$ the distributions $\nu_m^{N,p}$ and $\nu_m^p$ are close in the following sense (see Cui et al. (2023) for a similar setting).

**Lemma A.1** (Convergence of the measure). *Suppose Assumption 3.1 holds. Given the dynamics (14) and (16) for every $n = 0, \ldots, T$ it holds:*

$$\sup_{p\in\mathcal{P}} \mathbb{E}\left[\|\nu_n^{N,p} - \nu_n^p\|\right] = \mathcal{O}(1/\sqrt{N}). \tag{18}$$

*Proof.* We are going to follow an induction argument over the time steps:

*Initialization:* for $n = 0$, since we have indipendent samples at the starting point, by the law of large numbers (LLN) we have:

$$\sup_{p\in\mathcal{P}} \mathbb{E}\left[\|\nu_0^{N,p} - \nu_0^p\|\right] \to 0$$

with rate of convergence $\mathcal{O}\left(\frac{1}{\sqrt{N}}\right)$.

In particular, let us denote $\mathcal{S} := \{y_1, \cdots, y_K\}$, $\nu_0^p(y_i) = p_i$, $\nu_0^{N,p}(y_i) = \frac{1}{N}\sum_{i=1}^{N}\delta_{Y_i^\alpha}(y_i) = \frac{C(y_i)}{N}$, where $C(y_i)$ is defined as the number of agent that are in the state $y_i$ at time 0. We can write:

$$\mathbb{E}\left[\|\nu_0^{N,p} - \nu_0^p\|\right] = \frac{1}{2}\mathbb{E}\left[\sum_{i=1}^{|\mathcal{S}|}\left|\frac{C(y_i)}{N} - p_i\right|\right] \le \frac{\sqrt{|S|}}{2}\mathbb{E}\left[\sum_{i=1}^{|\mathcal{S}|}\left(\frac{C(y_i)}{N} - p_i\right)^2\right]^{1/2}$$

by Cauchy-Schwarz inequality. Notice now that $C(y_i) \sim Bin(N, p_i)$ and so

$$\sum_{i=1}^{|S|} Var\left(\frac{C(y_i)}{N}\right) = \sum_{i=1}^{|S|} \frac{p_i(1-p_i)}{N} = \frac{1 - \sum_{i=1}^{|S|} p_i^2}{N} \leq \frac{|S|-1}{N|S|}$$

since the quantity $1 - \sum_{i=1}^{|S|} p_i^2$ has its max when $p_i = \frac{1}{|S|}$.

Eventually we obtain the explicit constant:

$$\mathbb{E}\left[\|\nu_0^{N,p} - \nu_0^p\|\right] \leq \frac{\sqrt{|S|-1}}{2\sqrt{N}}.$$

*Remark* A.2. Notice that the bound depends on the cardinality of the state space: more states lead to a larger upper bound, meaning possibly a larger discrepancy between the empirical and mean field distributions. This is due to the fact that we used as metric the total variation distance, which sums over all possible states. In continuous space this metric is not feasible and so usually the Wasserstein distance is used for convergence analysis (see Carmona and Delarue (2018)). Actually in the finite space and discrete time setting we have the following inequality:

$$d_{min}\|\mu - \nu\|_{TV} \leq W_1(\mu, \nu) \leq D\|\mu - \nu\|_{TV},$$

where $d_{min} := \min_{x \neq y} d(x, y)$ and $D := \max_{x \neq y} d(x, y)$. Notice that the Wasserstein distance in finite space and discrete time is defined as:

$$W_p(\mu, \nu) = \left(\min_{T \in \mathcal{C}(\mu, \nu)} \sum_{i=1}^n \sum_{j=1}^n d(x_i, x_j)^p \cdot T_{i,j}\right)^{\frac{1}{p}},$$

where $\mathcal{C}(\mu, \nu)$ is the set of couplings defined as:

$$\mathcal{C}(\mu, \nu) = \left\{ T \in \mathbb{R}^{n \times n} \;\middle|\; \sum_{j=1}^n T_{i,j} = \mu_i \; \forall i, \; \sum_{i=1}^n T_{i,j} = \nu_j \; \forall j, \; T_{i,j} \geq 0 \; \forall i, j \right\},$$

and $d(x_i, x_j)$ is the distance between points $x_i$ and $x_j$ in the metric space. More details on Wasserstein distances are described in Villani (2009) and Arjovsky et al. (2017).

*Induction step:* assume now that (18) holds at time $n$. Using triangle inequality, at time $n+1$ we have, for any $p \in \mathcal{P}$,

$$\mathbb{E}\left[\|\nu_{n+1}^{N,p} - \nu_{n+1}^p\|\right] \leq$$

$$\leq \mathbb{E}\left[\|\nu_{n+1}^{N,p} - \bar{F}(\nu_n^{N,p}, p_n(\nu_n^{N,p}))\|\right] + \mathbb{E}\left[\|\bar{F}(\nu_n^{N,p}, p_n(\nu_n^{N,p})) - \nu_{n+1}^p\|\right]$$

where we recall the expression of $\bar{F}$ described by (8).

For the second term, by Lipschitz property of $\bar{F}$ and $p(\nu)$, we can write :

$$\mathbb{E}\left[\|\bar{F}(\nu_n^{N,p}, p_n(\nu_n^{N,p})) - \nu_{n+1}^p\|\right]$$

$$= \mathbb{E}\left[\|\bar{F}(\nu_n^{N,p}, p_n(\nu_n^{N,p})) - \bar{F}(\nu_n^p, p_n(\nu_n^p))\|\right]$$

$$\leq L_{\bar{F}}\mathbb{E}\left[\|\nu_n^{N,p} - \nu_n^p\| + \|p_n(\nu_n^{N,p}) - p_n(\nu_n^p)\|\right]$$

$$\leq L_{\bar{F}}\mathbb{E}\left[\|\nu_n^{N,p} - \nu_n^p\| + L_p\|\nu_n^{N,p} - \nu_n^p\|\right]$$

$$= (L_{\bar{F}}(1 + L_p))\mathbb{E}\left[\|\nu_n^{N,p} - \nu_n^p\|\right] \leq (L_{\bar{F}}(1 + L_p))^{n+1}\mathbb{E}[\|\nu_0^{N,p} - \nu_0^p\|]$$

$$\leq (L_{\bar{F}}(1 + L_p))^{n+1} \frac{\sqrt{|S|-1}}{2\sqrt{N}}$$

by induction step, and the upper bound is independent of $p \in \mathcal{P}$ (since the constant $L_p$ is the same for all the control $p \in \mathcal{P}$).

For the first term we have:

$$\mathbb{E}\left[\left\|\nu_{n+1}^{N,p} - \bar{F}(\nu_n^{N,p}, p_n(\nu_n^{N,p}))\right\|\right] =$$

$$= \mathbb{E}\left[\left\|\frac{1}{N}\sum_{i=1}^{N}\delta_{Y_{n+1}^{i,\alpha}} - \bar{F}(\nu_n^{N,p}, p_n(\nu_n^{N,p}))\right\|\right] =$$

$$= \frac{1}{2}\mathbb{E}\left[\left|\sum_{y\in\mathcal{S}}\frac{1}{N}\sum_{i=1}^{N}\delta_{Y_{n+1}^{i,\alpha}}(y) - \bar{F}(\nu_n^{N,p}, p_n(\nu_n^{N,p}))(y)\right|\right] =$$

$$= \frac{1}{2}\sum_{y\in\mathcal{S}}\mathbb{E}\left[\left|\frac{1}{N}\sum_{i=1}^{N}\delta_{Y_{n+1}^{i,\alpha}}(y) - \bar{F}(\nu_n^{N,p}, p_n(\nu_n^{N,p}))(y)\right|\right] =$$

$$= \frac{1}{2}\sum_{y\in\mathcal{S}}\mathbb{E}\left[\mathbb{E}\left[\left|\frac{1}{N}\sum_{i=1}^{N}\delta_{Y_{n+1}^{i,\alpha}}(y) - \bar{F}(\nu_n^{N,p}, p_n(\nu_n^{N,p}))(y)\right|\,\Big|\,\boldsymbol{Y_n^\alpha}\right]\right]$$

The interpretation of $\bar{F}$ gives us:

$$\bar{F}(\nu_n^{N,p}, p_n(\nu_n^{N,p}))(y) = \sum_{y'}\nu_n^{N,p}(y')\mathbb{P}(Y_{n+1}^p = y|Y_n^p = y')$$

$$= \frac{1}{N}\sum_{i=1}^{N}\mathbb{P}(Y_{n+1}^p = y|Y_n^p = Y_n^{i,p})$$

$$= \frac{1}{N}\sum_{i=1}^{N}\mathbb{P}(Y_{n+1}^{i,p} = y|Y_n^{i,p})$$

$$= \frac{1}{N}\sum_{i=1}^{N}\mathbb{E}[\delta_{Y_{n+1}^{i,p}}(y)|Y_n^{i,p}]$$

where we used that the $i$ particles are indistinguishable and have the same transition functions. So we can conclude the argument as:

$$\mathbb{E}\left[\left\|\nu_{n+1}^{N,p} - \bar{F}(\nu_n^{N,p}, p_n(\nu_n^{N,p}))\right\|\right] =$$

$$= \frac{1}{2}\sum_{y\in\mathcal{S}}\mathbb{E}\left[\mathbb{E}\left[\left|\frac{1}{N}\sum_{i=1}^{N}\delta_{Y_{n+1}^{i,\alpha}}(y) - \mathbb{E}\left[\frac{1}{N}\sum_{i=1}^{N}\delta_{Y_{n+1}^{i,\alpha}}(y)\Big|\boldsymbol{Y_n^\alpha}\right]\right|\,\Big|\,\boldsymbol{Y_n^\alpha}\right]\right] \le \frac{|S|}{4\sqrt{N}}$$

by the LLN, where again the bound is independent of $p \in \mathcal{P}$.

Indeed, given the past history $\boldsymbol{Y_n^\alpha}$ the random variables $\delta_{Y_{n+1}^{i,\alpha}}$ become conditionally independent for every $i = 1, \ldots, N$. Furthermore each $\delta_{Y_{n+1}^{i,\alpha}(y)}$ is a Bernoulli random variable, therefore its variance $Var(\delta_{Y_{n+1}^{i,\alpha}}(y)|\boldsymbol{Y_n^\alpha}) \le \frac{1}{4}$. Summing over all agents, the variance of the empirical mean becomes $\frac{1}{4N}$. Using Cauchy-Schwarz inequality, for any random variable $Z$ with finite variance $\mathbb{E}[|Z - \mathbb{E}[Z]|] \le \sqrt{Var(Z)}$, so in our case we obtained the constant $\frac{|S|}{4\sqrt{N}}$. We have thus proved by induction that:

$$\sup_{p\in\mathcal{P}}\mathbb{E}\left[\|\nu_n^{N,p} - \nu_n^p\|\right] \le \left[(L_{\bar{F}}(1+L_p))^{n+1}\frac{\sqrt{|S|-1}}{2} + \frac{|S|}{4}\right]\frac{1}{\sqrt{N}}$$

for every time step $n = 0, \ldots, T$. $\qquad\square$

This result allows us to prove the following main theorem on the optimal cost approximation in the $N$-agent problem. This is a precise version of the informal statement in Theorem 3.2.

**Theorem A.3** ($\varepsilon$-approximation of the $N$-agent problem)**.** *Suppose Assumption 3.1 holds. Given the dynamics (14) and (16) and the social cost associated (15), (17), let us denote by $p^*$ the optimal policy for the mean field problem and by $\hat{p}$ the optimal policy for the $N$-agent problem. It holds:*

$$J^N(p^*, \ldots, p^*) - J^N(\hat{p}, \ldots, \hat{p}) = \mathcal{O}(1/\sqrt{N}). \tag{19}$$

*Proof.* We can write:

$$J^N(p^*, \dots, p^*) - J^N(\hat{p}, \dots, \hat{p}) = \left(J^N(p^*, \dots, p^*) - J(p^*)\right) + \left(J(p^*) - J(\hat{p})\right) + \left(J(\hat{p}) - J^N(\hat{p})\right)$$

Notice first that we can bound this term simply deleting the second term in the r.h.s noticing $J(p^*) - J(\hat{p}) \leq 0$ since $p^*$ is optimal for the *mean field* cost $J(p)$. For the first term we can write:

$$J^N(p^*, \dots, p^*) - J(p^*)$$

$$= \mathbb{E}\left[\sum_{m=0}^{T} \Psi(\nu_m^{N,p^*}, p_m^*(\nu_m^{N,p^*}))\right] - \sum_{m=n}^{T} \Psi(\nu_m^{p^*}, p_m^*(\nu_m^{p^*}))$$

$$= \sum_{n=0}^{T} \mathbb{E}\left[\Psi(\nu_n^{N,p^*}, p_n^*(\nu_n^{N,p^*})) - \Psi(\nu_n^{p^*}, p_n^*(\nu_n^{p^*}))\right]$$

$$\leq L_\Psi \sum_{n=0}^{T} \mathbb{E}\left[\left\|\nu_n^{N,p^*} - \nu_n^{p^*}\right\| + \left\|p_n^*(\nu^{N,p^*}) - p_n^*(\nu_n^{p^*})\right\|\right]$$

$$\leq L_\Psi(1 + L_p) \sum_{n=0}^{T} \mathbb{E}\left[\left\|\nu_n^{N,p^*} - \nu_n^{p^*}\right\|\right]$$

$$\leq TL_\Psi(1 + L_p) \sup_{n \in \{0, \dots, T\}} \mathbb{E}\left[\left\|\nu_n^{N,p^*} - \nu_n^{p^*}\right\|\right] \leq TL_\Psi(1 + L_p)\left[(L_{\bar{F}}(1 + L_p))^T \frac{\sqrt{|S| - 1}}{2} + \frac{|S|}{4}\right]\frac{1}{\sqrt{N}},$$

by Lemma A.1. For the last term $J(\hat{p}) - J^N(\hat{p})$ we can apply the same argument that we just described. In the folliwng way we obtain:

$$J^N(p^*, \dots, p^*) - J^N(\hat{p}, \dots, \hat{p}) \leq 2TL_\Psi(1 + L_p)\left[(L_{\bar{F}}(1 + L_p))^T \frac{\sqrt{|S| - 1}}{2} + \frac{|S|}{4}\right]\frac{1}{\sqrt{N}}$$

$$\square$$

# B    PROOF OF THEOREM 4.1

Let us prove Theorem 4.1.

*Proof.* To prove this result, we will show that we can reduce the problem to a mean field optimal control problem in discrete time and continuous space. Then we can apply the well-studied dynamic programming principle for mean field Markov decision processes (MFMDPs) (see e.g. Motte and Pham (2022); Carmona et al. (2023); Bäuerle (2023)). We have:

$$V_n(\nu) = \inf_{p \in \mathcal{P}_{n,T}} \sum_{m=n}^{T} \sum_{(x,a) \in \mathcal{S}} \nu_m^{p,\nu,n}(x,a)\Phi(x, \mu_m^{p,\nu,n})ap_m(x)$$

$$= \inf_{p \in \mathcal{P}_{n,T}} \sum_{m=n}^{T} \Psi(\nu_m^{p,\nu,n}, p_m),$$

where $\Psi : \mathcal{P}(\mathcal{X} \times \{0,1\}) \times \mathcal{P}(\mathcal{X}) \to \mathbb{R}$ and it is defined as:

$$\Psi(\nu, q) := \sum_{(x,a) \in \mathcal{S}} \nu(x,a)\Phi(x, \nu_X)aq(x).$$

Then we can define the process $Z$ taking value in $\mathcal{P}(\mathcal{X} \times \{0,1\})$:

$$Z_n^p = z = \nu; \qquad Z_m^p := \nu_m^{p,\nu,n} \quad \forall m \geq n$$

such that it follows the dynamics $Z_{m+1}^p = \bar{F}(Z_m^p, p_m)$ for every $m = n, \dots, T-1$. We can write:

$$V_n(z) = \inf_{p \in \mathcal{P}_{n,T}} \sum_{m=n}^{T} \Psi(Z_m^p, p_m),$$

and we recognize a well studied control problem for which the DPP is:

$$V_n(z) = \inf_{h \in \mathcal{H}} \Psi(z, h) + V_{n+1}(\bar{F}(z, h)).$$

where $\mathcal{H}$ is the set of all functions $h : \mathcal{X} \to [0, 1]$. Finally we can recover our result:

$$V_n(\nu) = \inf_{h \in \mathcal{H}} \sum_{(x,a) \in \mathcal{S}} \nu(x, a) \Phi(x, \nu_X) a h(x) + V_{n+1}(\bar{F}(\nu, h)). \tag{20}$$

where $\nu_X$ is the first marginal of the distribution $\nu$. $\qquad\square$

## C  ALGORITHMS

Alg. 3 and 4 present respectively the direct approach and the DP-based method.

---

**Algorithm 3** Direct Approach for MFOS

---

**Require:** Time-dependent stopping decision neural network: $\psi_\theta : \{0, \ldots, T\} \times \mathcal{X} \times \mathcal{P}(\mathcal{S}) \to [0, 1]$, cost function $\Phi$, mean-field dynamic transition $\bar{F}$, time horizon $T$, max training iteration $N_{\texttt{iter}}$.
    **// TRAINING**
1: **for** $k = 0, \ldots, N_{\texttt{iter}} - 1$ **do**
2:     Uniformly sample initial distribution $\nu_0^p$ from the probability simplex on $\mathbb{R}^{2|\mathcal{X}|}$
3:     **for** $n = 0, \ldots, T$ **do**
4:         $p_n(x) = \psi_\theta(x, \nu_n^p, n; \theta_k)$ for any $x \in \mathcal{X}$          $\triangleright$ Compute stopping probability
5:         $\ell_n = \sum_{x \in \mathcal{X}} \nu_n^p(x, 1) \Phi(x, \mu_n) p_n(x)$          $\triangleright$ Compute loss at time $n$
6:         $\nu_{n+1}^p = \bar{F}(\nu_n^p, p_n)$          $\triangleright$ Simulate MF dynamic
7:     $\ell = \sum_{n=0}^{T} \ell_n$          $\triangleright$ Compute the total loss
8:     $\theta_{k+1} = \texttt{optimizer\_update}(\theta_k, \ell(\theta_k))$          $\triangleright$ AdamW optimizer step
9: Set $\theta^* = \theta_{N_{\texttt{iter}}}$
10: **return** $\psi_{\theta^*}$

---

**Algorithm 4** Dynamic Programming Approach for MFOS

---

**Require:** A sequence of stopping decision neural network: $\psi_\theta^n : \mathcal{X} \times \mathcal{P}(\mathcal{S}) \to [0, 1]$ for $n \in \{0, \ldots, T-1\}$, cost function $\Phi$, mean-field dynamic transition $\bar{F}$, time horizon $T$, max training iteration $N_{\texttt{iter}}$.
    **// TRAINING**
1: Set $\psi_\theta^T = 1$ since all distribution stopped at time $T$.
2: **for** $n = T - 1, \ldots, 0$ **do**          $\triangleright$ Train backward in time
3:     **for** $k = 0, \ldots, N_{\texttt{iter}} - 1$ **do**
4:         Uniformly sample initial distribution $\nu_n^p$ from the probability simplex on $\mathbb{R}^{2|\mathcal{X}|}$
5:         **for** $m = n, \ldots, T$ **do**
6:             **if** $m = n$ **then**
7:                 $p_m(x) = \psi_\theta^m(x, \nu_m^p; \theta_k^n)$          $\triangleright$ Compute with NN for current time
8:             **else**
9:                 $p_m(x) = \psi_\theta^m(x, \nu_m^p; \theta^{m,*})$          $\triangleright$ Compute with trained NN from future time
10:         $\ell_m = \sum_{x \in \mathcal{X}} \nu_m^p(x, 1) \Phi(x, \mu_m) p_m(x)$          $\triangleright$ Compute loss at time $m$
11:         $\nu_{m+1}^p = \bar{F}(\nu_m^p, p_m)$          $\triangleright$ Simulate MF dynamic
12:         $\ell = \sum_{m=n}^{T} \ell_m$          $\triangleright$ Compute the total loss from time $n$ to $T$
13:         $\theta_{k+1}^n = \texttt{optimizer\_update}(\theta_k^m, \ell(\theta_k^n))$          $\triangleright$ AdamW optimizer step
14:     Set $\theta^{n,*} = \theta_{N_{\texttt{iter}}}^n$          $\triangleright$ Stored trained weight

---

## D  IMPLEMENTATION DETAILS

In this section, we will discuss the choice of neural networks, training batch size, learning rate, and iterations, and all the related hyperparameters as well as computational resources used.

**Neural Network Architectures:** We have 4 variants of neural networks.

For the direct approach, the neural network takes an input time $t$, while for the DPP approach, the neural network does not need time input.

For the asynchronous stopping problem, besides time, the neural network has two spatial inputs 1) the state $x$, represented as an integer, goes through an embedding layer with learnable parameters and the results are fed to other operations. 2) the distribution $\nu$, represented as a vector, is inputted to the neural net directly. For the synchronous stopping problem, the neural network only has one spatial input, which is the distribution $\nu$, and is treated as the same way as discussed before.

In general, our neural network has the following structure. Our neural network takes an input pair $(x, t)$, where $x$ is the spatial input, $t$ is the time. If $t$ is a needed input, then it is passed through a module to generate a standard sinusoidal embedding and then fed to 2 fully connected layers with Sigmoid Linear Unit (SiLU) and generate an output $t_{\text{out}}$. Spatial input $x$ is passed through an MLP with $k$ residual blocks, each containing 4 linear layers with hidden dimension $D$ and SiLU activation. This generates an output $y_{\text{out}}$. Our final output out is computed through,

$$\text{out} = \text{Outmod}(\text{GroupNorm}(y_{\text{out}} + t_{\text{out}}))$$

where Outmod is an out module that consists of 3 fully connected layers with hidden dimension $D$ and SiLU activation, GroupNorm stands for group normalization. If $t$ is not a needed input, then set $t_{\text{out}} = 0$.

For all the test cases we have experimented with, we use $k = 3, D = 128$ for all the 1D experiments and $k = 5, D = 256$ for the 2D experiments.

**Computational Resources:** We run all the numerical experiments on an RTX 4090 GPU and a Macbook Pro with M2 Chip. For any of the test cases, one run took at most 3 minutes on GPU and 7 minutes on CPU.

**Training Hyperparameters:** For all the experiments, we choose an initial learning rate $10^{-4}$ of the AdamW optimizer. Each training is at most $10^4$ iterations, with a batch size 128. The number of training iterations is chosen based on numerical evidence and trial and error. We start with a moderate number and then increase it if the model shows signs of undertraining and is far from convergence.

## E   NUMERICAL EXPERIMENTS details

This section aims to complete the results of the 6 numerical experiments conducted. While some of the following plots have been previously discussed in Section 6, we provide the full descriptions of Example E.1 and Example E.2 here for the sake of completeness.

### E.1   EXAMPLE 1: TOWARDS THE UNIFORM

We take state space $\mathcal{X} = \{0, 1, 2, 3, 4\}$, time horizon $T = 4$, transition function $F(n, x, \mu, \epsilon) = x+1$ which means that the agent deterministically moves to the state on the right, with boundary at $x = 4$ (meaning that once at 4, the agent does not move anymore), and cost function $\Phi(x, \mu) = \mu(x)$ which depends on the mean field only through the state of the agent (this is sometimes called local dependence). For the testing distribution, we take a distribution concentrated on state $x = 0$, denoted as $\mu_0 = \delta_0$. It can be seen that the optimal strategy consists in spreading the mass to make it as close as uniform as possible (hence the name of this example). Fig. 8 shows that the testing loss decays towards the true optimal value, and the distribution evolves towards a uniform distribution as expected. Fig. 9 shows the losses with DPP: there is one curve per time step. At time 0, the value is close to the optimal value. First, we explain how the optimal value is computed. Since the agents move deterministically to the right, the only option to freeze some mass at a state $x$ is to do it at time $n$. It can be seen that: for every $n = 0, \ldots, T$ and for every $x \in \mathcal{X}$, we want to have $p_n(x = n) = \frac{1}{T+1-n} \mathbb{1}_{x=n}$ for $n < T$ and $p_n(x) = 1$ for $n = T$. Actually notice that for all $x \neq n$ the choice of $p_n$ is arbitrary so, at every time-step $n$ we can apply the same $p_n$ for every state $x$. This brings us to optimize over the set of synchronous stopping times.

Then we can compute the optimal value and obtain: $V^{*,\delta_0} := \frac{T+2}{2(T+1)}$.

Figs. 10 and 11 show the result for synchronous stopping.

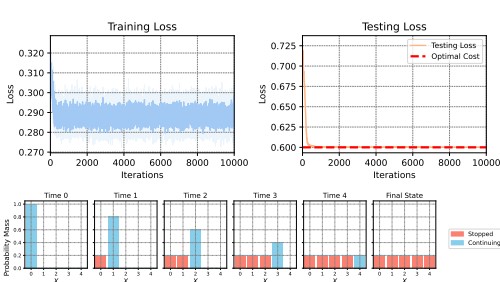

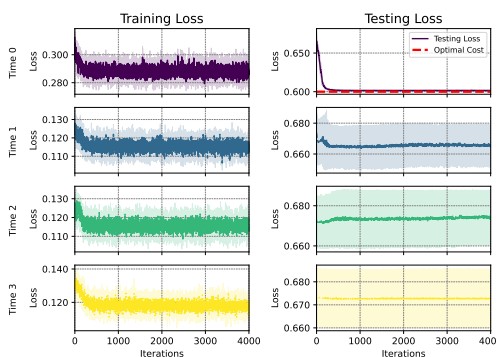

Figure 8: Example 1. DA results, asynchronous stopping. Top: training and testing losses. Bottom: evolution of the distribution after training.

Figure 9: Example 1. DPP results, asynchronous stopping. Training and testing losses.

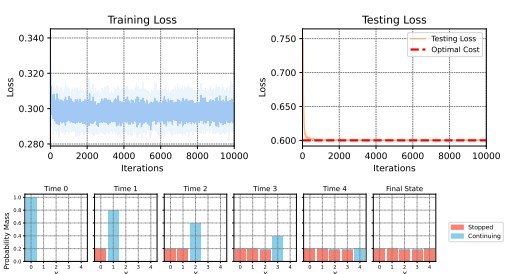

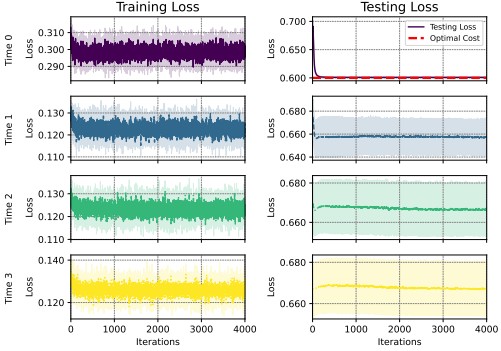

Figure 10: Example 1. DA results, synchronous stopping. Top: training and testing losses. Bottom: evolution of the distribution after training.

Figure 11: Example 1. DPP results, synchronous stopping. Training and testing losses.

### E.2 EXAMPLE 2: ROLLING A DIE

In this example, at every time step, a fair six-sided die is rolled. This takes the role of the noise $\epsilon \sim \mathcal{U}(\mathcal{X})$ where $\mathcal{X} = \{1, 2, 3, 4, 5, 6\}$. The system starts in the initial distribution $\eta = \frac{1}{4}\delta_1 + \frac{1}{4}\delta_2 + \frac{1}{2}\delta_5$, and evolves according to the dynamics (5) with: $\mu_0 = \eta$, $F(n, x, \mu, \epsilon_{n+1}) = \epsilon_{n+1}$. The social cost function associated to this scenario is $\Phi(x, \mu) = x$. DR and DPP results are shown in Figs. 12 and 13 respectively. Here again we observe convergence to the true optimal value. The optimal value is computed as follows. Using the dynamic programming principle described in (11) we can compute the optimal strategy and the optimal value:

$$p_0(\cdot) = (1, 0, 0, 0, 0, 0) \qquad p_1(\cdot) = (1, 1, 0, 0, 0, 0)$$
$$p_2(\cdot) = (1, 1, 0, 0, 0, 0) \qquad p_3(\cdot) = (1, 1, 0, 0, 0, 0)$$
$$p_4(\cdot) = (1, 1, 1, 0, 0, 0) \qquad p_5(\cdot) = (1, 1, 1, 1, 1, 1)$$

$$V^{*,\eta} = 1, 6525.$$

For our considered initial distribution, this is one of the possible optimal strategies, since we have no mass on some states and thus can assign any stopping probability to them. However, the solution we have presented is the only optimal solution for all possible initial distributions. Note that if we optimize on the class of synchronous stop times, we do not reach the same optimal value, but we reach a higher value, concluding that for this type of problem, it is better to optimize on asynchronous stop times. In fact, when you narrow the decision only to the class of synchronous stop times is better to

stop everyone at the first initial state reaching a value of $\tilde{V}* = 3, 25 > 1, 6525 = V^*$. Synchronous stopping results are shown in Fig. 14 and 15.

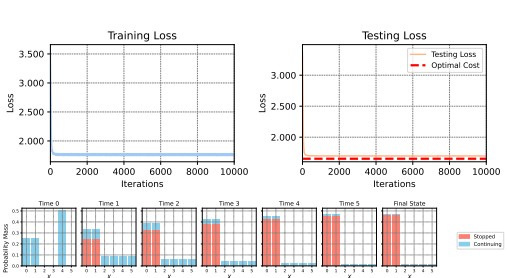

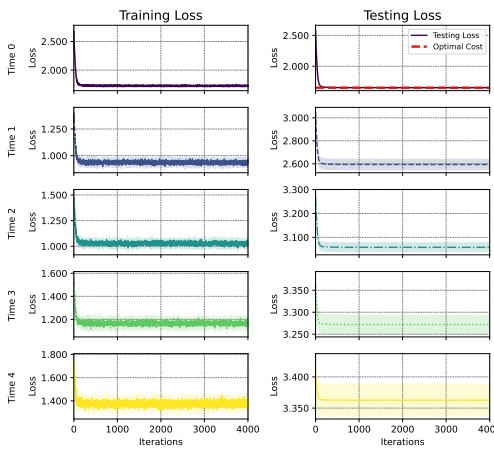

Figure 12: Example 2. DA results, asynchronous stopping. Top: training and testing losses. Bottom: evolution of the distribution after training.

Figure 13: Example 2. DPP results, asynchronous stopping. Training and testing losses.

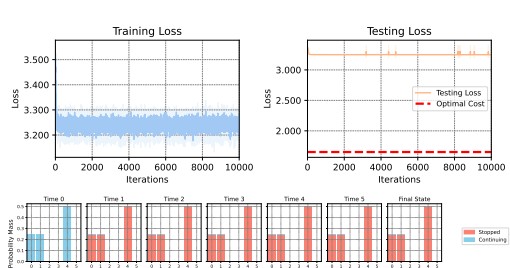

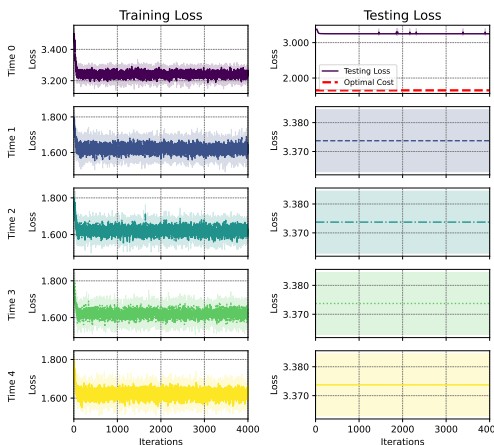

Figure 14: Example 2. DA results, synchronous stopping. Top: training and testing losses. Bottom: evolution of the distribution after training.

Figure 15: Example 2. DPP results, synchronous stopping. Training and testing losses.

### E.3 EXAMPLE 3: CROWD MOTION WITH CONGESTION.

This example extends the previous one, adding a congestion factor. The transition probabilities are:

$$p_n(z, x) := P(X_{n+1} = z | X_n = x) = \begin{cases} \frac{1}{6}(1 - \frac{1}{5}C_{\text{cong}}\mu(x)), & \text{if } z \neq x, \\ \frac{1}{6}(1 + C_{\text{cong}}\mu(x)), & \text{if } z = x. \end{cases} \quad (21)$$

Let us set $C_{\text{cong}} = 0.8$. However, the reasoning regarding the differences between scenarios in which the central planner optimizes the set of asynchronous stopping times or the set of synchronous stopping times is similar.

DPP testing and training losses are shown in Fig. 16.

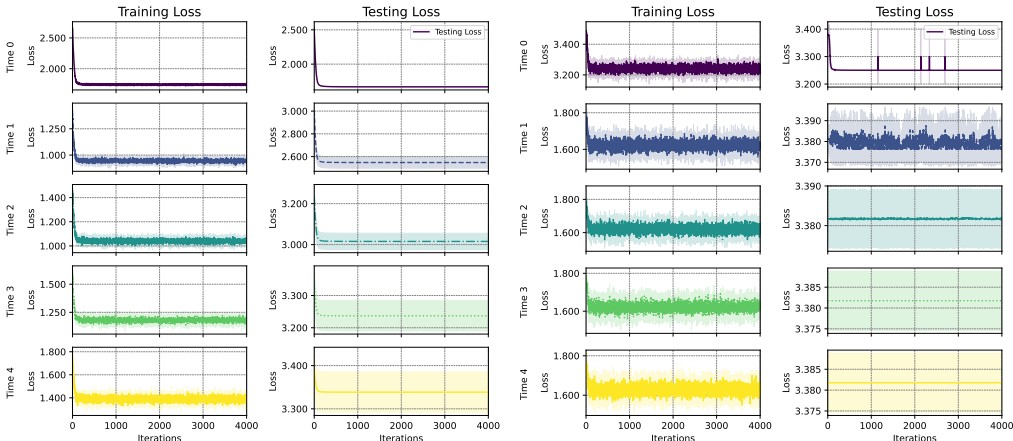

Figure 16: Example 3. DPP results. Training and testing losses. Left: asynchronous stopping. Right: synchronous stopping.

### E.4 EXAMPLE 4: DISTRIBUTIONAL COST

Synchronous stopping results are shown in Fig. 17.

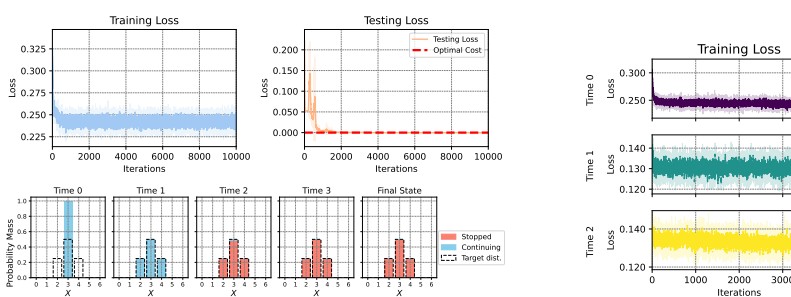 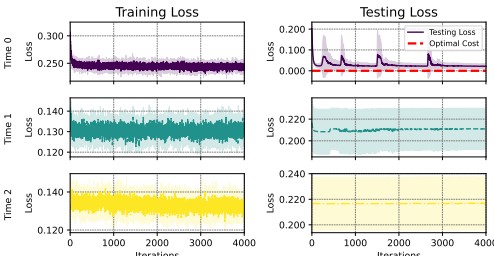

Figure 17: Example 4. DA results, synchronous stopping. Top: training and testing losses. Bottom: evolution of the distribution after training.

Figure 18: Example 4. DPP results, synchronous stopping. Training and testing losses.

### E.5 EXAMPLE 5: TOWARDS THE UNIFORM IN 2D

Asynchronous stopping results, including training losses, testing losses, distribution evolution, and stopping probability are shown in Figs. 19 and 20. Synchronous stopping results are shown in Figs. 21 and 22.

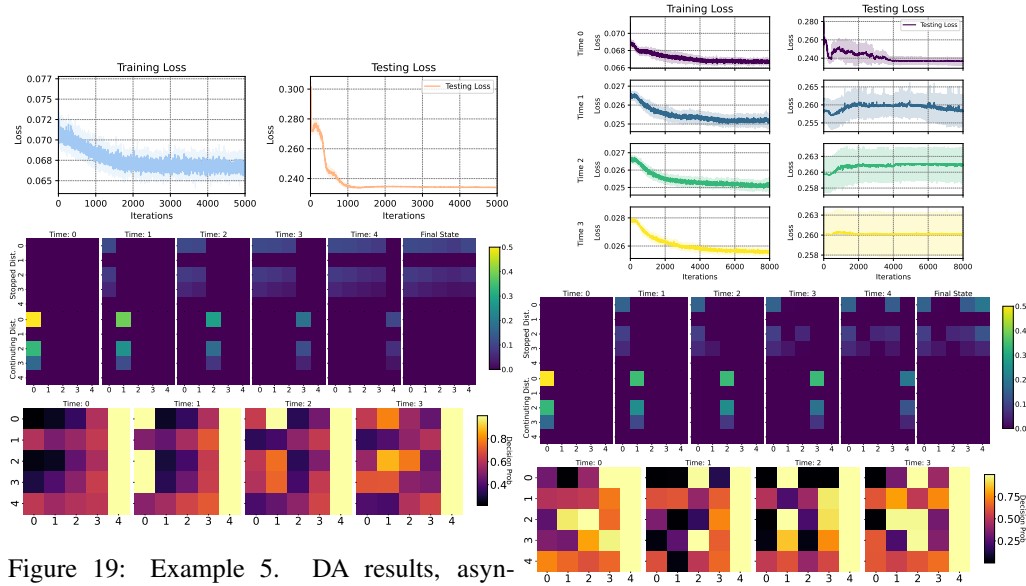

Figure 19: Example 5. DA results, asynchronous stopping. Top: training and testing losses. Bottom: evolution of the distribution and stopping probability after training.

Figure 20: Example 5. DPP results, asynchronous stopping. Top: training and testing losses. Bottom: evolution of the distribution and stopping probability after training.

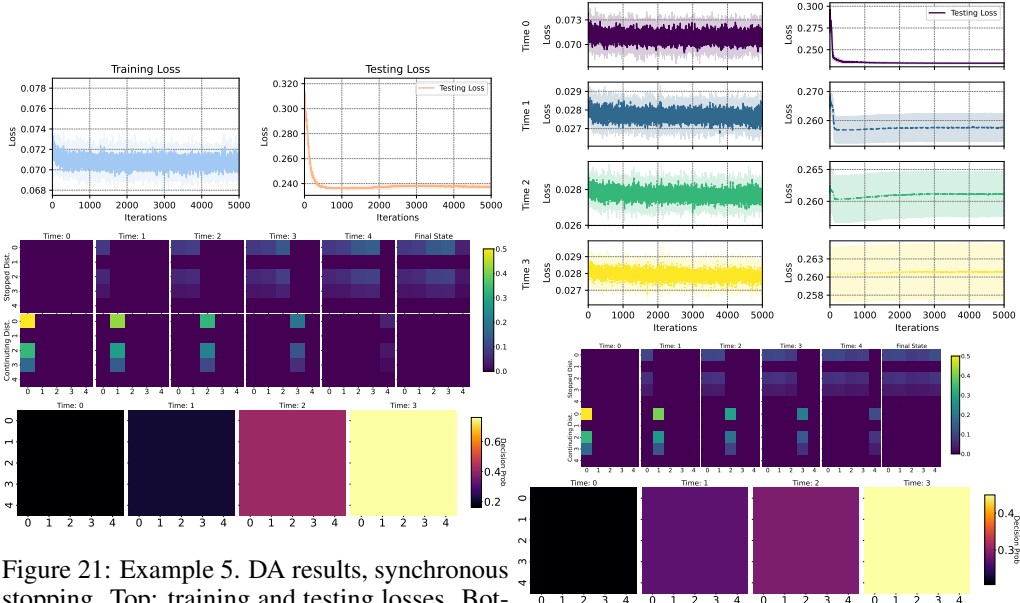

Figure 21: Example 5. DA results, synchronous stopping. Top: training and testing losses. Bottom: evolution of the distribution and stopping probability after training.

Figure 22: Example 5. DPP results, synchronous stopping. Top: training and testing losses. Bottom: evolution of the distribution and stopping probability after training.

### E.6  EXAMPLE 6: MATCHING A TARGET WITH A FLEET OF DRONES.

In this example, we extend our framework by incorporating a terminal cost and common noise. This allows us to consider a richer and more realistic class of MFOS environments. We extend the dynamics defined in (5) in the following way:

$$
\begin{cases}
X_0^\alpha \sim \mu_0, \qquad A_0^\alpha = 1 \\
\alpha_n \sim \pi(\cdot | X_n^\alpha) = Be(p_n(X_n^\alpha)) \\
A_{n+1}^\alpha = A_n^\alpha \cdot (1 - \alpha_n) \\
X_{n+1}^\alpha = \begin{cases} F(n, X_n^\alpha, \mu_n^\alpha, \epsilon_{n+1}, \epsilon_{n+1}^0), & \text{if } A_n^\alpha \cdot (1 - \alpha_n) = 1 \\ X_n^\alpha, & \text{otherwise.} \end{cases}
\end{cases}
\tag{22}
$$

where, $\epsilon_n^0$ is the common noise that affects the dynamics of all agents equally. Note that with the presence of common noise the mean field distribution $\nu$ is not deterministic, but it is a random variable that evolves conditionally with respect to the common noise.

Furthermore the social cost defined in (9) can be extended by adding a terminal cost:

$$
J(p) = \mathbb{E}^0 \left[ \sum_{n=0}^T \sum_{(x,a) \in \mathcal{S}} \left( \nu_n^p(x,a) \Phi(x, \nu_{X,n}^p) a p_n(x) \right) + g(\nu_{X,T}^p) \right],
\tag{23}
$$

where $g : \mathcal{P}(\mathcal{X}) \to \mathbb{R}$ is the terminal cost and $\mathbb{E}^0$ is the expectation with respect the common noise realization.

The results for DA for different target distributions are provided in Fig. 23. The results for DPP for different target distributions are provided in Fig. 24.

It is evident that, unlike the DPP, the optimal strategy in the DA tends to stop with high probability at the final time steps, as clearly illustrated for the target distributions corresponding to the letters "O" and "S".

## F  HYPERPARAMETERS SWEEP

In this section, we show the results of a sweep over the learning rate for Example 1 with the two methods and the two types of stopping times. We consider learning rates $10^{-2}$, $10^{-3}$ and $10^{-4}$ in this order in the plots from top to bottom.

Direct method stopping: Figs. 25 and 26 show the losses for the asynchronous and the synchronous stopping times respectively.

Direct method stopping: Figs. 27 and 28 show the losses for the asynchronous and the synchronous stopping times respectively.

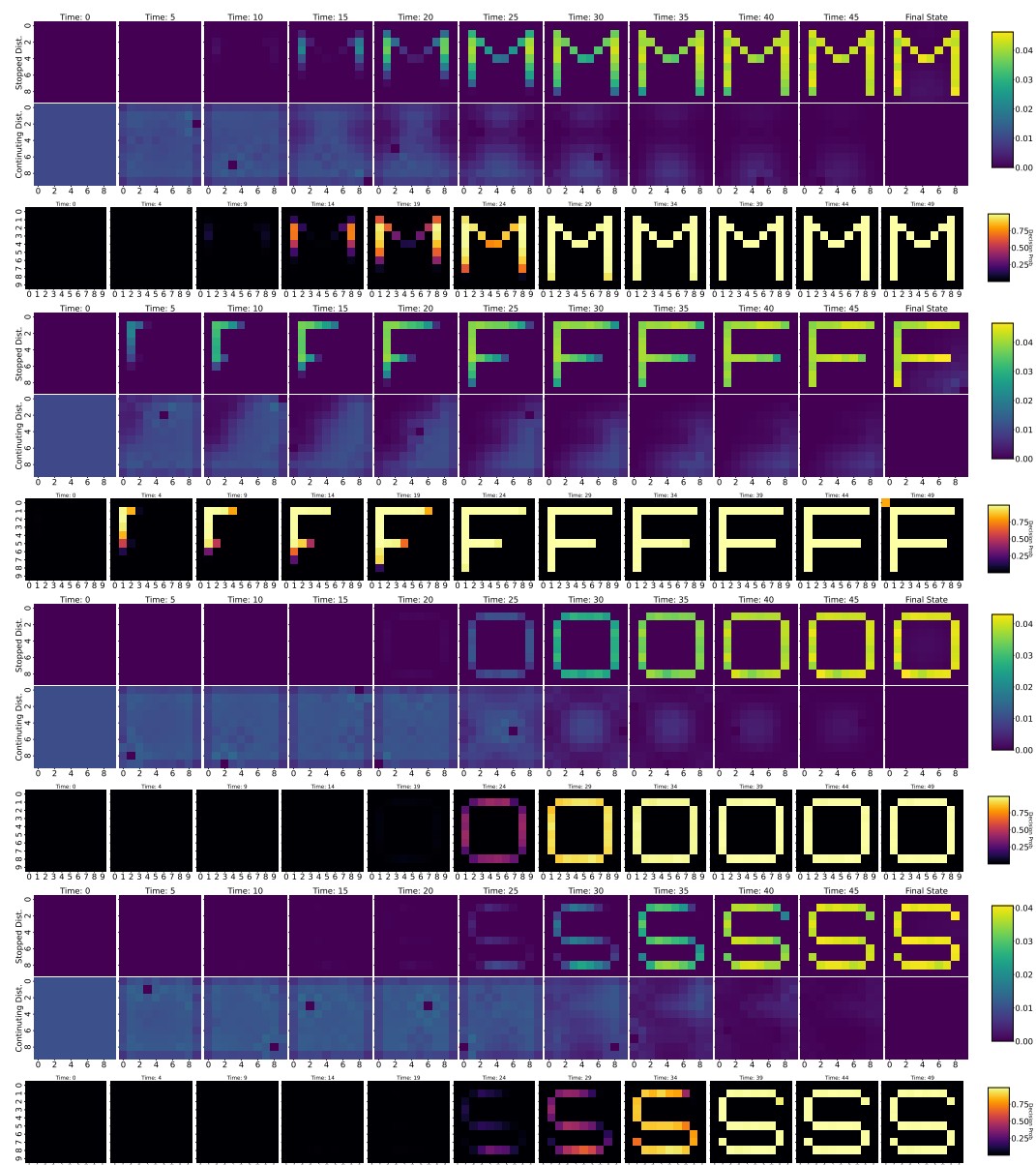

Figure 23: Example 6. DA results, asynchronous stopping. Match the Letter "M", "F", "O", "S", in a $10 \times 10$ grid with common noise.

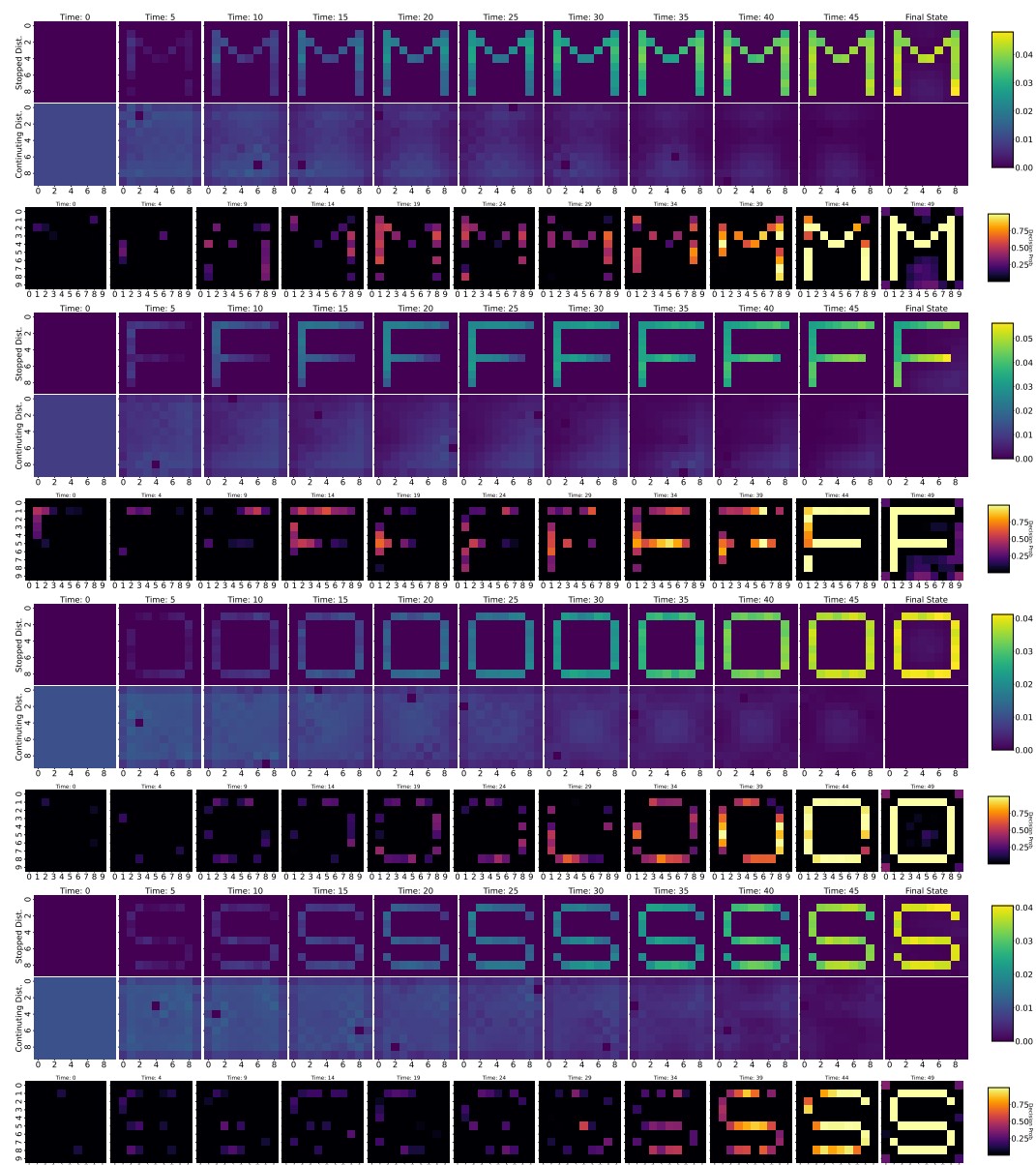

Figure 24: Example 6. DPP results, asynchronous stopping. Match the Letter "M", "F", "O", "S", in a $10 \times 10$ grid with common noise

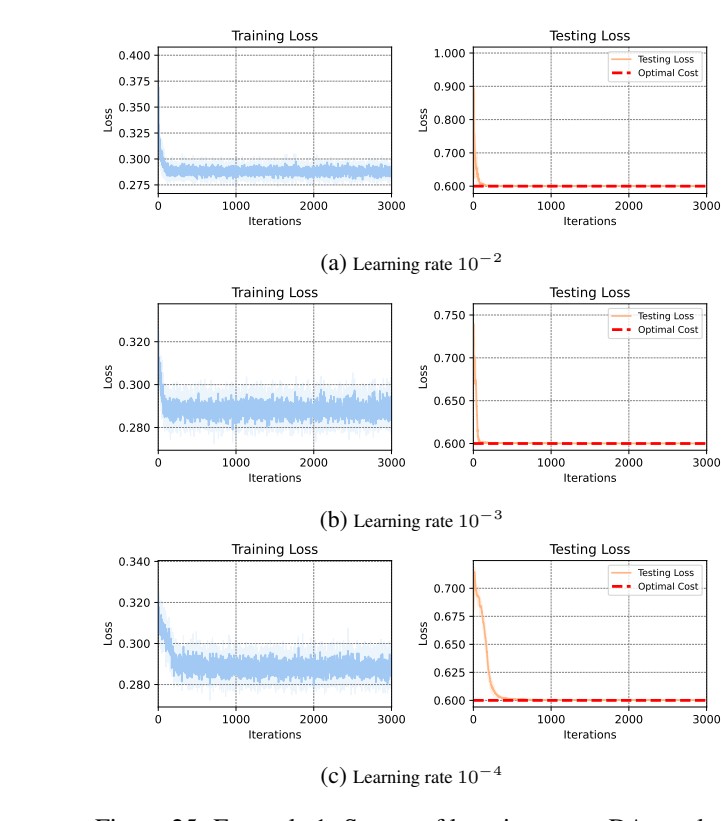

Figure 25: Example 1: Sweep of learning rates. DA results, asynchronous stopping.

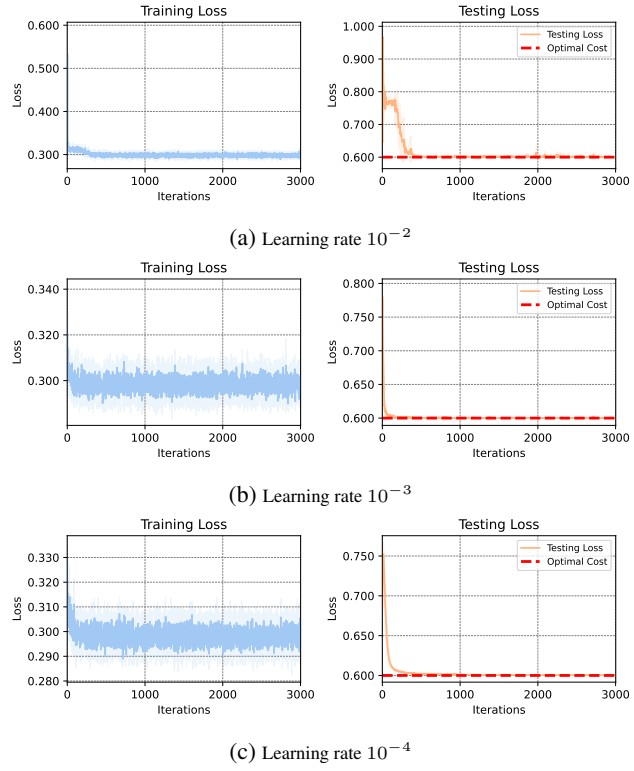

Figure 26: Example 1: Sweep of learning rates. DA results, synchronous stopping.

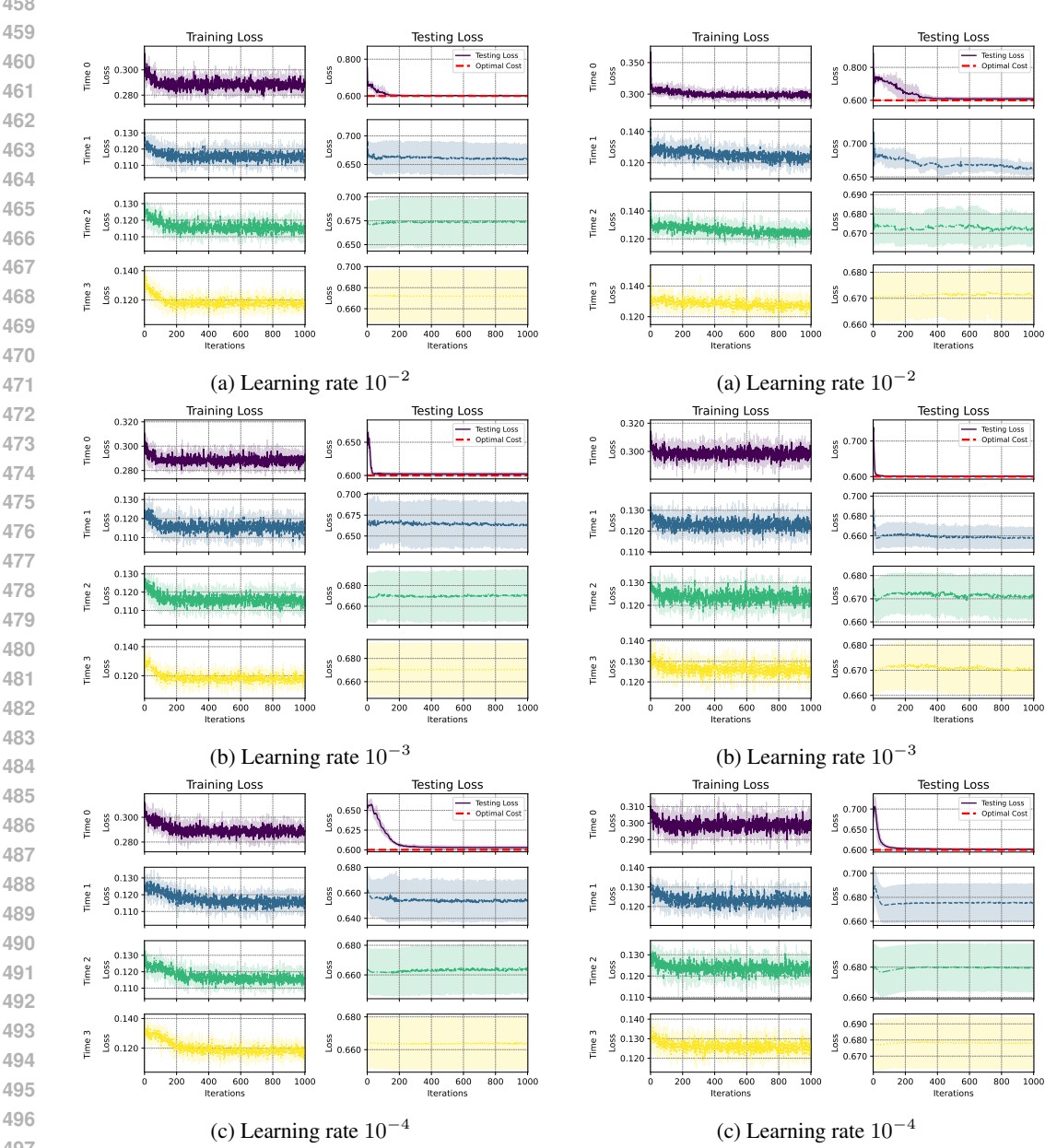

(a) Learning rate $10^{-2}$     (a) Learning rate $10^{-2}$

(b) Learning rate $10^{-3}$     (b) Learning rate $10^{-3}$

(c) Learning rate $10^{-4}$     (c) Learning rate $10^{-4}$

Figure 27: Example 1: Sweep of learning rates. DPP results, asynchronous stopping.

Figure 28: Example 1: Sweep of learning rates. DPP results, synchronous stopping.

