# OpenReview forum: "Deep Learning Algorithms for Mean Field Optimal Stopping in Finite Space and Discrete Time"
_ICLR.cc/2025/Conference — Submitted to ICLR 2025_

### Official Review · Reviewer_ijgf · 2024-11-04

**Soundness:** 2
**Presentation:** 3
**Contribution:** 2
**Rating:** 5
**Confidence:** 3

**Summary:**

This paper studies the mean field optimal stopping problem in finite space and discrete time. Theoretically, the authors show that as the number of agents N goes to infinity, the optimal cost converges to its mean field limit at a rate of $N^{-1/2}$. They also show the dynamic programming principle in this setting. Numerically, the authors propose two algorithms, one directly optimizes the objective, and the other leverages the maximum principle. The authors apply the algorithms on 6 examples to test the efficacy.

**Strengths:**

The numerical experiments are comprehensive, showing the effectiveness of the algorithm.

**Weaknesses:**

Part of the article is not clear enough. I have put details in Questions.
The notations in the paper are complicated and hard to track. For example, stopping corresponds to $\alpha=1$ and $A=0$, while $\alpha=0$ and $A=1$ refer to continuing the dynamic. This causes some confusion.

**Questions:**

Page 3 line 156. You mention that “randomized stopping times for individual agents differ from the randomization of the central planner on policies”. Why are they different? What should a central planner be like?

Page 4. What is the space for a stopping probability p? Is it a sequence of T functions in x, with values in [0,1]?

In assumption 2.1, you assume the Lipschitz condition of $\bar{F}$ and $\Psi$. However, you did not specify the metric in the input domain, which makes the assumption ambiguous.

---

> ### Author Response · Authors · 2024-11-19
> **Response to Reviewer ijgf**
>
> Thank you for your comments. We provide detailed answers below. If further clarifications are needed, please feel free to ask. Otherwise, we hope that you will consider raising your score.
>
> **Weakness (notations):**
> We are sorry that the notations are quite complex. We made some efforts to clarify some of them in the revised version. For $A$ and $a$ specifically, our rationale was that: $A$ stands for "Alive" while $\alpha$ stands for "action" (which is to stop or not). So $A=1$ means the agent has not stopped yet; when the agent stops, $\alpha=1$, and $A$ switches to $0$.  We have **added a sentence** to clarify this on lines 196-198.  One advantage of using this convention is that we can write the cost as in Eq. (6); otherwise, we would have to replace all the $\alpha$'s by $(1-\alpha)$. We are open to discussing other notations to ensure our paper is as clear and concise as possible.
>
> **Questions 1 (individual vs common randomization):**
> Thank you for your question. This comment is not central to our paper so **we removed the expression** *"and differs from the randomization of the central planner on policies"* to avoid any confusion. But let us explain better here. The central planner chooses a policy, which is a function $p$, where $p_n(s,\nu)$ gives the stopping probability for an agent when her state is $s$, the time step is $n$, and the mean field is $\nu$. So the agent will sample a random variable for herself to decide to stop or not by using a Bernoulli random variable with parameter $p_n(s,\nu)$. We call this randomness the *"individual randomization"*. Our comment meant to say that (in the whole paper) $p$ is a deterministic function, and the randomness is only at the individual level. One could also consider that *$p$ itself is a random function*, which would be interpreted as randomness at the central planner's level (or *"common randomization"*). But this would add an extra layer of complexity without bringing any clear way to find better policies.
>
> **Questions 2 (definition of $p$):**
> By using the notation $p:\{0,\dots,T\}\times\mathcal{X} \to[0,1]$, we mean that we can view $p$ as a function of time  and space taking values in $[0,1]$. Alternatively, it can be viewed as a sequence $(p_n)_{n=0,\dots,T}$. For every $n \in  \{0,\dots,T\}$ and $x \in \mathcal{X}$, $p_n(x) = p(n,x)$ is interpreted as the probability to stop at time $n$ in the state $x$. In other words, $p$ is the parameter of the Bernoulli distribution according to which the agent will choose to stop or not. When $p_n(x)$ is larger, agents in $x$ have higher probability of stopping.
>
>
> **Questions 3 (Lipschitz condition):** Thank you for raising this technical point. Actually, since the objects are finite-dimensional, all the norms are equivalent. But to be specific, we use the **total variation distance** (see Appendix A.2 line 733, when we use the definition of the metric). Notice that the space we are working on has finite dimension $2|\mathcal X|$ where $\mathcal X$ is the state space.  In the new Remark A.2, we stressed the fact that in finite dimensional spaces it is equivalent to the 2-Wasserstein distance which is the natural distance used in continuous spaces, see e.g., Carmona and Delarue (2018).
>
> **Based on these responses to your valuable feedback, we hope you will consider increasing your score for our paper.**

---

> > ### Author Response · Authors · 2024-12-02
> > **Discussion with Reviewer ijgf**
> >
> > We hope that you found our answers helpful. If there is anything else we can add to convince you that our paper deserves to be accepted, please feel free to let us know. Thank you for your time and attention.

---

### Official Review · Reviewer_v9ZM · 2024-11-04

**Soundness:** 3
**Presentation:** 4
**Contribution:** 3
**Rating:** 6
**Confidence:** 3

**Summary:**

This paper studies a special form of mean field reinforcement learning (RL), the mean-field optimal stopping problem (MFOS), where each agent can only choose to continue or stop. The authors prove that MFOS is a good approximation of the multi-agent optimal stopping problem (MAOS) and characterize the convergence rate as the number of agents tends to infinity. They also propose two algorithms for optimizing the policy in MFOS. The authors verify the effectiveness of their proposed algorithms in several numerical simulations.

**Strengths:**

The paper is well-structured and easy to follow because the authors provide sufficient background and motivations for the MFOS. The convergence rate concerning the number of agents (Theorem 2.2) should be a good contribution if it is the first theoretical result that characterizes the gap between multi-agent RL and mean field RL. However, I’m unsure if similar results exist in the literature on mean field RL. The proposed algorithms are natural and show a good empirical performance.

**Weaknesses:**

Problem setting: MFOS is a special class of mean field RL. The readers may wonder why this work's theoretical guarantees and algorithms cannot directly apply to general transition kernels and action sets.

Theoretical results: The $1/\sqrt{N}$ convergence rate in Theorem 2.2 is not fast enough if we consider the practical number of agents in target applications (e.g., the number of drones in a fleet in Example 6). But it is good to see that this convergence rate is tight.

Algorithms: A major limitation is that both algorithms, Direct Approach (DA) and Dynamic Programming (DP), need to know the exact mean-field dynamics $\bar{F}$ and take derivatives of this function.

Besides, the authors mention that DP is "A more ideal treatment" than DA in Line 325. However, I do not see any rigorous reasoning through time/memory complexity. The only practical reason might be the training memory size mentioned in Line 374. But even when T is large, it may be better to have a single time-dependent neural network (as required by DA) rather than training T in different neural networks (as needed by DP).

Experiments: The examples are all about synthetic environments with known transition probabilities.
Despite the above limitations, I still feel this work makes contributions that should be interesting to the audience in the field of multi-agent RL.

**Questions:**

I hope the authors can clarify whether similar convergence results like Theorem 2.2 exist in the literature of mean-field RL. If not, does the proof of Theorem 2.2 rely on the special structures of MAOS compared with general multi-agent RL?

Is the mean-field dynamics easy to learn/differentiate in real-world applications with a finite number of agents? Are the proposed algorithms robust to any estimation errors in the mean-field dynamics?

---

> ### Author Response · Authors · 2024-11-19
> **Response to Reviewer v9ZM (Weaknesses)**
>
> We thank you for the supportive review, and for highlighting our work as well-structured and demonstrating strong empirical performance. We have addressed each of the points in detail, aiming to clarify any doubts and justify a higher score. If further clarifications are needed, we would appreciate further references regarding the weaknesses mentioned.
>
> **Weakness 1 (MFOS vs MFRL):**
> Thank you for pointing out a possible connection with "mean field RL" but could you please provide a specific reference please? We are not sure what you mean by *"MFOS is a special class of mean field RL"*. Please note that our paper does not discuss reinforcement learning (RL) at all. Furthemore, in this paper we consider a *cooperative* problem, so methods for *non-cooperative* multi-agent settings (e.g., Nash equilibria) are not relevant for our problem.
>
> **Weakness 2 (Rate of convergence):** We provided more information in point 4 of the common reply but please let us know if you would like further clarifications.
>
>
> **Weakness 3 (Model knowledge):**
> Thank you for pointing out that it would be interesting to investigate model-free methods. We stress that in this work, we propose the **first computational methods** for mean field optimal stopping. With this in mind, we  focus on situations where the model is known as stated in line 099.  Model-free methods will be explored in future works. However, please evaluate our paper in the context of deep learning methods for optimal control and optimal stopping. For example, (Becker et al., 2019) for optimal stopping problems, and (Han et al., 2018; Huré et al., 2021) for optimal control are **assuming that the model is known** and have received considerable attention from the optimal control and machine learning communities.
>
>
> **Weakness 4 (Comparison between algorithms):** Thank you for this interesting question. We provided more information in point 3 of our common reply. In addition, we want to add the following. As far as the **memory required for the parameters** of the neural network (NN), we agree with you that *"it may be better to have a single time-dependent neural network (as required by DA)".* However, DA requires **differentiating through the whole trajectory** during backpropagation, the memory required for this increases with $T$. (This is not just backpropagation through the NN layers, which is well optimized. It is backpropagation through the entire state's trajectory.) On the other hand, the DP approach proceeds step-by-step, requiring backpropagation through one time step only, with memory requirement independent of $T$.
>
>
> **Weakness 5 (Extensions to MARL):**
> Thank you for your positive comment. This paper is a first step towards efficient computational methods for large-scale MAOS. While we assume full knowledge of the model for now, we hope our work will attract more attention to this problem and will be used as a basis for model-free RL methods in the near future. Our examples are new and have been designed for this family of MAOS problems. They can hopefully be used as benchmark problems by other authors in future works.

---

> > ### Comment · Reviewer_v9ZM · 2024-11-26
> > **Thanks for your response**
> >
> > 1. I mean the mean-field game setting in [1], where each player faces a traditional MDP. In OS, each player only has two actions: "continue" or "stop." Cooperation can be achieved if every player shares the same reward function.
> > 2. Thank you for the clarification.
> > 3. Thank you for acknowledging this point. I raised this question based on my understanding of the connection between MFOS and MFRL. People often assume the transition probabilities are unknown in the RL literature (e.g., [1]). Thus, you can emphasize some unique challenges of the MFOS setting in this work.
> > 4-5. Thank you for the clarifications.
> >
> > [1] Laurière, Mathieu, et al. "Scalable deep reinforcement learning algorithms for mean field games." International Conference on Machine Learning. PMLR, 2022.

---

> > > ### Author Response · Authors · 2024-11-28
> > > **Reply to Reviewer v9ZM**
> > >
> > > Thank you for taking the time to provide us additional feedback. We reply to points 1 and 3:
> > >
> > > 1. It seems there is some confusion.
> > >     - On the one hand, we agree that a scenario with multiple agents who can stop their dynamics can be recast in the language of MDPs with 2 actions (continue/stop) and an state space which includes the agent's state and her status (continuing/stopped). This is also true at the mean-field level. In that sense, we can say that the multi-agent stopping **setting** is a special case of the multi-agent MDP setting.
> > >     - On the other hand, we disagree with your original claim: *"MFOS is a special class of mean field RL."*. This is because the **solution concept** is different. Specifically, we read in detail reference [1] (Lauriere et al.). It proposes two algorithms for solving Mean Field Games (MFG). That paper is about computing a **Nash equilibrium** while we focus on computing a **social optimum**. We do not see concretely how to implement in [1] the idea that "every player shares the same reward function". So their methods do not apply at all to our problem.
> > >
> > > We will clarify the conceptual connection in the revised version but we hope you agree that the existence of [1] does not jeopardize in any way our contributions.
> > >
> > > 3. Thanks for the suggestion. We will add explanations about unique challenges in our setting such as the fact that the problem is very high-dimensional due to the fact that the value function is a function of the whole population distribution and not just of the individual state (in contrast with e.g. [1]).
> > >
> > > [1] Laurière, Mathieu, et al. "Scalable deep reinforcement learning algorithms for mean field games." International Conference on Machine Learning. PMLR, 2022.

---

> > > > ### Comment · Reviewer_v9ZM · 2024-11-29
> > > > **Thanks for your response**
> > > >
> > > > I apologize for the confusion about Nash equilibrium and social optimum. The cooperative setting is indeed not a particular case of the competitive setting. However, I disagree that the solution concept is the main barrier to applying methods from the literature of mean-field RL. For example, in Chapter 4 of [2], the author defined a mean-field MARL setting where the agents cooperate to maximize the averaged reward. The author of [2] also proposed learning algorithms that does not require knowing the transition probabilities.
> > > >
> > > > I understand it is already close to the end of the discussion period. I will keep the score, and I hope my suggestions are constructive for any revision.
> > > >
> > > > [2] Gu, Haotian. Mean-Field Cooperative Multi-agent Reinforcement Learning: Modelling, Theory, and Algorithms. University of California, Berkeley, 2023.

---

> > > > > ### Author Response · Authors · 2024-12-02
> > > > > **Reply**
> > > > >
> > > > > Thank you for your response and the suggestions. As we mentioned earlier, in this work we assume that the model is known. We will study model-free reinforcement learning methods in future works.

---

> ### Author Response · Authors · 2024-11-19
> **Response to Reviewer v9ZM (Questions & References)**
>
> **Question 1 (Existing results on rate of convergence):**
> Similar results indeed exist in the literature on mean field control (MFC).  For example Corollary 1 in (Cui et al., 2024), but it does not provide a rate of convergence while we do. Theorem 2.1 of (Motte and Pham, 2022) is also in a similar spirit to our problem. However they consider infinite horizon discounted MFC while we focus on finite-horizon MFOS. So these existing results do not apply to our problem and our proof is tailor-made. Notice that it gives not only a rate of convergence but even the constant in the big $\mathcal{O}$ is explicit (see the proof of Theorem A.3 in Appendix).
>
> **Question 2 (Robust approximation):** Thank you for raising these two points:
> 1. "*Is the mean-field dynamics easy to learn/differentiate in real-world applications with a finite number of agents?*": For the type of problems we consider here, the interactions between agents are through the empirical distribution (see Eqs. (1)-(2)). So the computational complexity is related to how the dynamics and cost functions depend on this distribution. About **learning**: it would be interesting to investigate the question of learning the model, as has been done for mean field control e.g. in (Pásztor et al., 2023) (these results do not apply directly to our setting though). Because MFOS is close to (but not same as) MFC, this paper gives hope that we can indeed learn the model efficiently in our problems as well. About **differentiating**: with our approach, the input is the mean field $\mu^N_n$ and not the vector of positions for all the agents $(X^1_n,\dots,X^N_n)$. So we expect that the complexity only scales with the dimension of the mean field (i.e., the number of states) and not with the number of agents. For the complexity with respect to the number of agents, see also our reply to Question 1 of Reviewer SfDE.
>
> 2. "*Are the proposed algorithms robust to any estimation errors in the mean-field dynamics?*" Thank you for raising this interesting question. Although we did not address this question explicitly, the techniques used to prove Theorem 2.2 show that the MAOS problem is robust to a perturbation of the distribution. Furthermore, our Example 6 shows that even when the **distribution is perturbed by a common noise**, the algorithms still perform very well. Rigorously proving the stability of deep neural network algorithms is very challenging and is left for future work.
>
> **We hope that these clarifications will convince you to support our paper and raise your score.**
>
> **New references:**
>
> Pásztor, B., Krause, A., & Bogunovic, I. . Efficient Model-Based Multi-Agent Mean-Field Reinforcement Learning. Transactions on Machine Learning Research (2023).
>
> Cui, K., Hauck, S. H., Fabian, C., & Koeppl, H. Partially Observable Multi-Agent Reinforcement Learning using Mean Field Control. In ICML 2024 Workshop: Aligning Reinforcement Learning Experimentalists and Theorists.
>
> Han, J., Jentzen, A., & E, W. . Solving high-dimensional partial differential equations using deep learning. Proceedings of the National Academy of Sciences, 115(34), 8505-8510 (2018).
>
> Huré, C., Pham, H., Bachouch, A., & Langrené, N. Deep neural networks algorithms for stochastic control problems on finite horizon: convergence analysis. SIAM Journal on Numerical Analysis, 59(1), 525-557 (2021).

---

### Official Review · Reviewer_7hVt · 2024-11-09

**Soundness:** 3
**Presentation:** 3
**Contribution:** 3
**Rating:** 8
**Confidence:** 3

**Summary:**

This papers designs a mean field approximation method for solving multi-agent optimal stopping (MAOS). Theoretical justifications are provided establishing that the mean field approximation error is relatively small as number of agents goes to infinity. A dynamic programming principle is also established. Two algorithms are designed with neural networks modeling the policy functions. Adequate numerical experiments are provided.

**Strengths:**

1. The paper is overall clear and well-written.
2. The idea is innovative.
3. Solid theoretical results are shown with practical interpretations.
4. Adequate numerical experiments are conducted for the proposed algorithms.

**Weaknesses:**

I agree with the limitations discussed by the authors in Sec 6. Other than that, having some specific problem as an example in the first few sections could further improve readibility.

**Questions:**

1. Is the intuition of Theorem 2.2 similar to Law of Large Numbers, in the sense that averaging the stop/continue of N agents starts to loose randomness as N gets larger? And the rate of 1 / \sqrt{n} could roughly be related to the rate of concentration for the average?
2. What is the structure of neural networks for the applications? I.e. some shallow fully connected layers, or are some specific architectures prefered?

---

> ### Author Response · Authors · 2024-11-19
> **Response to Reviewer 7hVt**
>
> We thank you for recognizing the novelty of our work and its solid theoretical foundations. We are uncertain about why the initial score of 8 was changed to 6 and we hope that you will remain supportive of our work and will consider reverting the score to 8. We have addressed all the comments with detailed explanations below.
>
> **Weakness:**
> Thank you for the suggestion. We have added a motivating example **(Distributional Cost)**, at the end of Section 2.1. We believe it is important to introduce a simpler, yet equally interesting, example in the early sections to familiarize the reader with the topic. Note that this example is later solved in Section 6 as Example 4. For further details on the example, please refer to our previous response.
>
> **Question 1 (Propagation of chaos):** The intuition of Theorem 2.2 is based on the so-called "propagation of chaos". We gave explanations in the first paragraph of Section 2. We can also give the following comparison with LLN. **LLN** can be used only at time $0$ because $(X^i_0)_{i=1,\dots,N}$ are independent, and only to approximate an expected value by an empirical average. However, it cannot be applied at future times (because the particles are interacting) and it cannot be used to compute more complex functions of the population distribution. **Propagation of chaos** allows us to do that based on the intuition that, when $N$ goes to infinity, particle trajectories become decoupled because they only see the population distribution (which is deterministic in the limit $N=\infty$). To be specific, propagation of chaos yields that the empirical distribution of the $N$ particles converges to the state distribution of the limiting process. For more information, see e.g., Sznitman (1991). However, notice that we **do not use** an existing propagation-of-chaos result. We prove it completely, with arguments tailored to our problem, which allows us to get an explicit rate of convergence in Theorem 2.2 (even the constant in the big $\mathcal{O}$ is explicit, see the proof of Theorem A.3 in Appendix).
>
>
> **Question 2 (Architecture):** Thank you for the question. In fact, we described in details the neural network architectures in Appendix D, in paragraph **"Neural Network Architectures"**. We summarize here the main points. For all the numerical experiments conducted in this work, we use an MLP with residual connections and a varying hidden dimension. We processed the inputs to neural nets to acquire an embedding vector of same dimension that represents the information. We embed time using a fixed sinusoid vector, embed state with a learnable embedding table and embed mean field distribution vector with a learnable MLP. We found that this architecture is the most efficient among the ones we tried, although our algorithm can work with various other architectures. However, we do not think that shallow neural networks (e.g., with just one hidden layer) are enough for solving the numerical problems considered in this work, particularly for the more complex examples.
>
> **We appreciate your support and we hope that these answers (and the ones given to other reviewers) will convince you to raise your score.**
>
> **New reference:**
>
> Sznitman, A. S. (1991). Topics in propagation of chaos. Ecole d’été de probabilités de Saint-Flour XIX—1989, 1464, 165-251.

---

> > ### Comment · Reviewer_7hVt · 2024-11-26
> >
> > Thank you for your detailed response. Part of my questions and concerns have been addressed. I have raised my score back to 8.

---

### Official Review · Reviewer_SfDE · 2024-11-10

**Soundness:** 3
**Presentation:** 2
**Contribution:** 2
**Rating:** 5
**Confidence:** 3

**Summary:**

This work focuses on optimal stopping of stochastic processes in the multi-agent setting.
Prior work on optimal stopping:
1. provided deep learning approaches in the single-agent case,
2. extended the optimal stopping problem in the multi-agent case, referred to as multi-agent optimal stopping (MAOS).

Furthermore, as the MAOS problem complexity increases with the number of agents $N$, prior work:

3. has combined MAOS with mean-field approximation where $N\to\infty$ in the continuous time and continuous space setting.


This work builds on these approaches in that it considers a mean-field approximation for MAOS as in (2,3), but aims to develop computational methods as in (1).

It introduces two deep learning methods to efficiently find solutions, proving the approach efficient with experiments on high-dimensional problems.

**Strengths:**

- Proposes two deep learning methods that leverage trajectory simulation and backward induction for optimal stopping.
- Empirical validation: demonstrates the effectiveness of the methods through experiments on high-dimensional problems

**Weaknesses:**

## Lacking explanation of the motivation
The introduction does not give a motivating multi-agent example or explain examples of what kind of problems this work will allow to solve.
I also did not understand by reading the introduction why the results (and deep learning approaches) from the single-agent setting do not extend to the multi-agent setting. The introduction should emphasize the main challenges for the extension.

## Writing

*Title*. This paper is about a multi-agent setting extension, and the title does not even mention that.

*Abstract*.
- starting with "optimal stopping of stoch. processes" would be clearer, because there are other studied stopping criteria
- I couldn't tell what the DPP proof implies. Perhaps describe what it implies for MFOS without assuming the reader knows DPP and its implications.

*Introduction*. The problem of the paper is very vague while reading the introduction. It would be helpful to provide a simple multi-agent optimal stopping motivating example so that ML readers can understand for which type of problems this problem is relevant.


*Other.*
- In the listed contributions it is unclear if (1) is for the discrete or the continuous case.

### Structure
The results are not in a separate section, making it hard to understand the contributions vs. the background. Section 2 contains existing models and subsection 2.4 gives one of the main theorems.

### Minor
- line 100: the text in brackets is unclear, do you mean the net's input can be up to 300? How was this determined?
- lines 108-109. At this point, only $N$ is introduced, but you use $\delta, X, k, \alpha$, etc. Moreover, this notation cannot be found later in the paper.
- lines 116-118: It is weird that you first define  $P(\mathcal{X})$ and later $\mathcal{X}$.
- line 125: $\delta$ is not defined.
- line 126: typo mathds-$N$ should be $N$?
- line 133: missing 'is' (which is)
- line 139: write what is the expectation over in eq. (2)

**Questions:**

1. Regarding prior MAOS work, precisely how does the problem complexity increase with N?
2. From the two proposed Deep Learning approaches, which one is better (or in which case which one is better)?
3. Could you elaborate on how the deep learning approaches for discrete-time single-agent case OS problems (listed in line 84) differ from the approach taken here on a per-agent basis?
4. I do not understand what you mean by "this makes the problem time inconsistent" in line 170. I understand that Markovian is comp. efficient, but do not understand why otherwise is "time inconsistent".
5. How does the extended state in Eq. (5) differ from Talbi et al. (2023)?
6. Since Eq. (5) is for a representative player, how does it differ from the singe-agent setting? Why is the most complex step in extending the single-agent results to the multi-agent setting? (with mean field)

---

> ### Author Response · Authors · 2024-11-19
> **Response to Reviewer SfDE (Weaknesses)**
>
> Many thanks for your detailed review, which helped improve the quality of our paper. We have addressed all the points with detailed responses. Based on this, we hope you will consider raising your score.
>
> **Weakness 1 (motivations):** Thank you for giving us the opportunity to clarify the motivations for MAOS. Please see points (1) and (2) in our common reply, and let us know if you would like more details.
>
> **Weakness 2 (writing):**
> * *Title*: Thank you for this suggestion. We made the choice of focusing on "mean field", but we could change it, for example to "Deep Learning Algorithms for Multi-Agent Mean Field Optimal Stopping in Finite Space and Discrete Time". If changing the title would help to increase your score, please let us know and we will do it.
> * *Abstract:*
> 	* We added "stochastic processes" to avoid any confusion.
> 	* We rephrased how the DP method works: DPP gives a way to compute the value function and the stopping rule using backward induction.
> * *Contributions*: it is for the discrete problem; we updated the draft.
>
> **Weakness 3 (structure):**
> Thank you for the observation. Our main contributions are listed on page 2 (blue box). But in the revision, we follow your suggestion and we keep Sections 2.1 to 2.3 in Section 2 and then put Section 2.4 as a new Section 3.
>
> **Weakness 4 (minor):**
> Thank you for raising these points. We corrected them in the revised paper. We provide explanations for some of the observations:
> * Dimension, line 101 (former line 100): The problem dimension is explained in a dedicated paragraph at the beginning of Section 6, former Section 5 ("Problem Dimensions"): We count the total dimension of the neural network's input, which is $|\mathcal{X}| + 2 |\mathcal{X}| = 3|\mathcal{X}|$, where $|\mathcal{X}|$ is the dimension of an individual agent's state space.
> *  $\mathbb{N}$, line 127 (former line 126): In the definition of $F: \mathbb{N} \times\mathcal X\times\mathcal P(\mathcal X)\times E\to\mathcal X$, $\mathbb N$ represent the set of integers because the time variable is an integer. It is not the same as $N$, which is the number of agents.

---

> ### Author Response · Authors · 2024-11-19
> **Response to Reviewer SfDE (Questions)**
>
> **Questions 1 (complexity in N):**
> We are not aware of any results about the exact complexity of MAOS. However, we note that existing references only tackled models with a very small number of agents. For instance Best et al. (2018) has only up to 8 robots. Furthermore, there is a connection with optimal multiple stopping problems, in which the goal is to choose $N$ distinct stopping times. These problems are simpler than ours because there are no interacting agents. We note that for instance Han et al. (2023) solve the problem with $N$ only up to $3$. Last, to solve MAOS without using mean-field approximations, there are 2 issues:
> 1. The stopping decision should be a function of the vector of all the agents' states $(X^1_n,\dots,X^N_n)$, whose dimension increases linearly with the number $N$ of agents. We would expect computational complexity at least polynomial in $N$.
> 2. At every time step, one needs to find the optimal subset of agents to stop (see e.g., Eq. (3.3) in (Talbi et al., 2022)). But the number of subsets is exponential in the number of agents.
>
> In the mean field limit, these issues are avoided.
>
> **Question 2 (comparison of algos):** Please see point 2 in our common reply.
>
> **Question 3 (single vs multi-agent):**
> We provide some background information in point 1 of our common reply. To be more specific, let us compare with the method of Becker et al. (2019). First, they do not consider at all a direct approach like our Algo. 1. The algo. of Becker et al. is based on DPP, like our Algo. 2, but in a very different form. The only common point is that the value function is computed backward in time. But they optimize the stopping decision for a single agent, while we optimize for a whole population, hence solving the MFOS problem. trying to apply their method in our problem would lead to one of the two issues described in point 1 of our common reply. We hope this answers your question. If not, could you please clarify what you mean by *"on a per agent basis"*?
>
> **Question 4 (time inconsistency):**
> We use this term to mean that the problem, in its initial formulation (Eq. (3)), is not Markovian and therefore does not satisfy the Bellman optimality principle. Specifically, the transition from time $n$ to $n+1$ does not depend only on $X^\alpha_n$ but also on $\mu^\alpha_n$. In particular, agents that have stopped in the past remain stopped. This information is not included in the state $X^\alpha_n$ and hence its evolution is not Markovian and we cannot derive a DPP based purely on $X^\alpha_n$. Our use of the word time inconsistent is in line with other works, such as (D. Andersson & B. Djehiche, 2010) which stated "*the cost functional is also of mean-field type, which makes the control problem **time inconsistent** in the sense that the Bellman optimality principle does not hold"*.
>
>
> **Question 5 (extended state):**
> As mentioned in Section 2.3 (line 208) "*The idea of extending the state using the extra information is similar to Talbi et al. (2023*)". However, our Eq. (5) is in discrete time and space while their evolution is in continuous time and space (see Eq. (2.7) in the aforementioned paper). It implies that the methods are completely different since they choose to rely on partial differential equations, which do not make sense in discrete problems.
>
> **Question 6 (role of mean field in Eq. (5)):**
> In Eq. (5), the evolution of the representative player state $X_n^\alpha$ is given by  $F(n, X_n^{{\alpha}}, \mathcal{L}(X_n^\alpha), \epsilon_{n+1})$, which involves the law $\mathcal{L}(X_n^\alpha)$ of the random variable $X^\alpha_n$. So the state's evolution involves the distribution of the state itself. If we remove this term $\mathcal{L}(X_n^\alpha)$, then the problem reduces to a standard optimal stopping problem for one single agent. However, the presence of this term means that the agent cannot find an optimal stopping time without taking into consideration $\mathcal{L}(X_n^\alpha)$. When transitioning from the single-agent case to the multi-agent case, the crucial step involves accounting for the interactions between an agent and the rest of the population.
>
> **In light of these clarifications, would you consider increasing your score? If not, could you please let us know which additional clarifications are needed?**
>
> **New references:**
>
> Han, Y., & Li, N. A new deep neural network algorithm for multiple stopping with applications in options pricing. Communications in Nonlinear Science and Numerical Simulation, 117, 106881 (2023).
>
> Andersson, D., Djehiche, B. A Maximum Principle for SDEs of Mean-Field Type. Appl Math Optim 63, 341–356 (2011).
>
> Germain, M., Mikael, J. & Warin, X. Numerical Resolution of McKean-Vlasov FBSDEs Using Neural Networks. Methodol Comput Appl Probab 24, 2557–2586 (2022).
>
> Huré, C., Pham, H., Bachouch, A., Langrené, N.  Deep neural networks algorithms for stochastic control problems on finite horizon: convergence analysis. SIAM J Numer Anal 59(1):525–557 (2021).

---

> > ### Comment · Reviewer_SfDE · 2024-11-26
> > **Thanks for your response**
> >
> > Thank you for your detailed response and clarifications.
> >
> > I recommend revising the introduction and contributions section to more clearly emphasize the work accomplished, the novelty of the approach, and the technical challenges involved in integrating these elements.
> >
> > ---
> > After reading the responses and the remaining reviews, I have decided to keep my score due to the limited novelty and technical contributions (as outlined in my summary), as well as the significant revisions and presentation issues. I hope the feedback provided proves helpful to the authors.

---

### Author Response · Authors · 2024-11-19
**Common Reply**

We thank all the reviewers for their thoughtful and constructive feedback, which recognizes the strengths and innovative aspects of our work. We have addressed every comment in detail, aiming to clarify any ambiguities and reinforce the contributions of our manuscript. We hope these responses effectively resolve any concerns and demonstrate the value of our work, encouraging an improvement of the scores.

Before providing answers to each reviewer, we address a few key points in this common reply:

**1. Single vs Multi-Agent OS**: Optimal stopping (OS) focuses on a single-agent problem, and it is very different from multi-agent optimal stopping (MAOS). Indeed,
- In the single-agent case, the dynamics and the costs depend only on the individual agent's state; the stopping rule also depends only on that state.
- In MAOS, the dynamics and costs of agent $i$ involve the states of other agents, which creates an interdependence in the stopping decisions; the stopping rules are functions of the states of all the agents.
- We can envision two ways to use single-agent OS numerical methods for MAOS, but none of them is satisfactory:
	- First, we can view the whole system, consisting of the $N$ agents' states, as a single state, and apply OS methods for that state. Then the whole group has to stop at once. However, we want to allow agents to stop at different times.
	- Second, we can fix the behavior of $N-1$ agents, and solve the OS problem for the remaining agent. We need to update every agent iteratively in this way. But it is not clear why this would converge, and it is more likely to converge to a Nash equilibrium than a social optimum, as we want.

**2. Applications and examples:** Let us give more concrete motivations:
1. MAOS has gained significant importance in a variety of fields. Two specific applications are:
	- **Robotics:** in mission monitoring tasks, multiple mobile robots must observe the progress of another robot performing a specific task (Best et al., 2018).
	- **Finance:** pricing options with multiple stopping times (Kobylansky et al. 2011) can be viewed as an MAOS problem (see Talbi et al. (2024)).
2. To help understand MAOS, let us explain the **$N$-agent version of our numerical Example 4**:
	- Here, $F(n,x,\mu,\epsilon) = x+\epsilon$, which means that the next state of an agent is the previous state plus some random disturbance. Based on Eq. (1), the dynamics of agent $i$ is: $X_{n+1}^{i} = X_{n}^{i} + \epsilon_{n+1}^{i}$ if the agent does not stop, and $X_{n+1}^{i} = X_{n}^{i}$ otherwise.
	- The agents' states yield the empirical distribution $\mu^N_n$ (proportion of agents in each state).
	- Based on Eq. (2), if agent $i$ stops at time $n$, then she is incurred the cost: $\Phi(X^i_n,\mu^N_n) = \sum_{x} |\mu^N_n(x) - \rho_{\mathrm{target}}(x)|^2$, which is smaller if the agent stops when the population distribution matches the target one.
	- Some agents might have to stop even though the target distribution is not matched, so that others can later have a lower cost. This is a cooperative task to minimize the social cost.
	- Solving exactly the problem (finding the optimal stopping rule for every agent) is very complex. Our approach is to consider the mean-field problem which leads to an efficient solution (see Example 4, Section 6). Based on Theorem 2.2, the mean field solution provides a good approximation of the $N$-agent problem.

**3. DA vs DP algorithms:** We propose 2 algorithms, each with their own advantages. We refer to the dedicated paragraph "**Comparison of the Two Proposed Algorithms**" at the beginning of Section 6 (former Section 5). Based on empirical evidence,
- When computational resources are not a limitation, the stopping decision is learnt faster by the DA method than by the DP method.
- However the required memory for training with DA increases with $T$, whereas DP requires only constant order memory that is independent of the time horizon. This is because DA requires backpropagating gradients through the whole trajectory. So DP is more memory efficient when $T$ is large.
- Similar observations were made in continuous time optimal control by Germain et al. (2022); their local and global methods are analogous to our DP and DA methods respectively.

**4. Rate of convergence:** Theorem 2.2 justifies that the MFOS solution gives a good approximation of the MAOS problem, not only with an asymptotic result but with a rate of convergence of $\mathcal O(1/\sqrt{N})$. We want to stress that:
- It is possible that for some specific dynamics and costs with simple dependence on the mean field distribution, better rates can be obtained.
- However, under general conditions his result is expected to be optimal due to state-of-the-art results established in the mean field control literature (e.g., (Daudin et al. 2024).

For more details about each reviewer's points, please see our individual replies. We hope that these clarifications will convince the reviewers to increase their score.

---

> ### Author Response · Authors · 2024-11-19
> **Main Changes in Paper and References**
>
> We summarize here the main changes in the updated draft. We improved clarity and provided more motivations.
>
> To make it easier for reviewers to track the updates, **the changes made in the draft are highlighted in purple.**
> * The "Introduction" and the "Related Work" paragraph have been updated.
> * Line 110, we added a reference to (Sznitman, 1991) for background on the propagation of chaos.
> * We added a "Motivating Example" at the end of Section 2.1 (Distributional Cost).
> * To increase clarity, we added blue boxes for the main contributions (p.2) and the 3 main theorems; we also added a paragraph containing the major challenges addressed by this work at the end of Section 2.1 (red box).
> * To avoid confusion, we removed the comment about common randomization in Section 2.2.
> * We added a sentence in Section 2.3 to clarify the notations.
> * We moved Section 2.4 in a separate section (new Section 3).
> * The conclusion has been updated.
> * We moved to Appendix E.1 and E.2 the brief descriptions of Example 1 and Example 2 respectively, due to space constraints.
> * We placed the details of the transition probabilities of Example 3 in Appendix E.3.
> * We added a comment on the metric that has been used in the proofs (total variation distance) in line 733.
> * We corrected the typos pointed out by reviewer SfDE.
>
> **References**  for the common reply:
>
> Daudin, S., Delarue, F., & Jackson, J. On the optimal rate for the convergence problem in mean field control. Journal of Functional Analysis, 287(12), 110660, (2024).
>
> Germain, M., Mikael, J., & Warin, X. Numerical resolution of McKean-Vlasov FBSDEs using neural networks. Methodology and Computing in Applied Probability, 24(4), 2557-2586, (2022).

---

### Meta-Review · Area_Chair_51tV · 2024-12-19

**Metareview:**

This paper considers the mean-field optimal stopping (MFOS) problem, a multi-agent version of the optimal stopping in the limit where the number of agents becomes very large, so the agents are essentially playing against a mean field. The authors provide an approximation theorem connecting the mean-field approximation to large-but-finite numbers of agents, as well as a dynamic programming principle (DPP) that characterizes the mean-field solution. Finally, they provide two algorithms for solving the MFOS problem based on deep learning (but with no theoretical guarantees).

Most of the reviewers were on the fence about this paper, for several reasons:
- First, the paper is difficult to read, and the positioning of its contributions is somewhat unclear. The authors' comments and revisions did help in this regard, but did not fully alleviate the problem.
- Another concern was that the proposed algorithms require global knowledge of the exact mean-field dynamics, as captured by $\bar F$ (including being able to differentiate it). This assumption is conceptually ill-suited to the multi-agent setting, which the authors motivate as agents interacting and stopping on their own (that is, based on local algorithms / rules).

To be sure, the paper isn't without its merits. At the same time however, it was not possible to make a clear case for acceptance of the paper "as is" (that is, without considerable further revisions requiring a fresh look), and the paper was not championed during the discussion phase. As a result, it was decided to make a "reject" recommendation, while encouraging the authors to take into account the feedback provided by the reviewers and resubmit at the next opportunity.

**Additional Comments On Reviewer Discussion:**

In view of the limitations mentioned above, it was not possible to make a clear case for acceptance. The paper was not championed by any of the reviewers during the discussion phase, and the conclusion was to encourage the authors to take into account the feedback provided by the reviewers and recommend to resubmit at the next opportunity.

---

### Decision · Program_Chairs · 2025-01-22

Reject